# Diverse calcium dynamics underlie place field formation in hippocampal CA1 pyramidal cells

Mate Sumegi[1], Gaspar Olah[1], Istvan Paul Lukacs[1], Martin Blazsek[2,3], Judit K Makara[2]*, Zoltan Nusser[1]*

[1]Laboratory of Cellular Neurophysiology, HUN-REN Institute of Experimental Medicine, Budapest, Hungary; [2]Laboratory of Neuronal Signaling, HUN-REN Institute of Experimental Medicine, Budapest, Hungary; [3]Roska Tamás Doctoral School of Sciences and Technology, Faculty of Information Technology and Bionics, Pázmány Péter Catholic University, Budapest, Hungary

## eLife Assessment

This **fundamental** study provides new insights into the plasticity mechanisms underlying the formation of spatial maps in the hippocampus. Supported by a large and comprehensive dataset, the evidence is **convincing**. This study will be of interest to neuroscientists focusing on spatial navigation, learning, and memory.

*For correspondence:
makara.judit@koki.hu (JKM);
nusser.zoltan@koki.hu (ZN)

Competing interest: The authors declare that no competing interests exist.

**Abstract** Every explored environment is represented in the hippocampus by the activity of distinct populations of pyramidal cells (PCs) that typically fire at specific locations called their place fields (PFs). New PFs are constantly born even in familiar surroundings (during representational drift), and many rapidly emerge when the animal explores a new or altered environment (during global or partial remapping). Behavioral time scale synaptic plasticity (BTSP), a plasticity mechanism based on prolonged somatic action potential (AP) bursts induced by dendritic $Ca^{2+}$/NMDA plateau potentials, was recently proposed as the main cellular mechanism underlying new PF formations (PFFs), but it is unclear whether burst-associated large somatic $[Ca^{2+}]$ transients are always necessary and/or sufficient for PFF. To address this issue, here we performed in vivo two-photon $[Ca^{2+}]$ imaging of hippocampal CA1 PCs in head-restrained mice to investigate somatic $[Ca^{2+}]$ dynamics underlying PFFs in familiar and novel virtual environments. Our results demonstrate that although many PFs are formed by BTSP-like events, PFs also emerge with initial $[Ca^{2+}]$ dynamics that do not match any of the characteristics of BTSP. BTSP- and non-BTSP-like new PFFs occur spontaneously in familiar environments, during neuronal representational switches, and instantaneously in new environments. Our data also reveal that solitary $[Ca^{2+}]$ transients with larger amplitudes than those evoking BTSP-like PFFs frequently occur without inducing PFs, demonstrating that large $[Ca^{2+}]$ transients per se are not sufficient for PFF.

## Introduction

The hippocampus plays a critical role in spatial and contextual memory processes by forming cognitive maps of environments (*O'Keefe and Dostrovsky, 1971*; *Moser et al., 2008*; *Lisman et al., 2017*) that can support effective navigation toward learned salient locations (*Robinson et al., 2020*). During navigation, many pyramidal cells (PCs) in the dorsal hippocampal CA1 area (CA1PCs) exhibit action potential (AP) firing that is restricted to spatial locations, the cell's place fields (PFs). In different

environments, the activity of CA1PCs reorganizes to collectively form distinct representations, that is creating specific maps of each environment. In a novel environment, many CA1PCs are active already during the first traversal of their PFs, but a fraction of cells become place cells after some delay (*Hill, 1978*; *Wilson and McNaughton, 1993*; *Frank et al., 2004*; *Epsztein et al., 2011*; *Cohen et al., 2017*; *Sheffield et al., 2017*; *Dong et al., 2021*). Furthermore, spatial maps also gradually reorganize during continuous or repeated exploration of already familiar environments due to changes in the activity of individual PCs, including the recruitment of new place cells into the representation (*Mankin et al., 2012*; *Ziv et al., 2013*; *Rubin et al., 2015*; *Dong et al., 2021*; *Keinath et al., 2022*). Despite the fundamental importance of the dynamic hippocampal population code in spatial navigation, the mechanisms initiating and mediating new PF formations (PFFs) are still poorly understood.

How can place cells acquire their PFs? While cells that are already active at the first traversal of their PFs may simply receive sufficient active spatially tuned synaptic inputs *ab ovo* (*Epsztein et al., 2011*; *Cohen et al., 2017*), delayed PFF is assumed to require dynamic changes in synaptic inputs or the intrinsic excitability of CA1PCs. The cellular mechanisms underlying de novo formation of PFs have been elusive due to difficulties in measuring the subthreshold activity pattern that drives AP firing. However, recent studies have shed light on possible mechanisms. Most strikingly, in vivo patch-clamp recordings in head-fixed mice running on a familiar cue-rich treadmill revealed the spontaneous appearance of new PFs in previously silent cells, following the sudden occurrence of a robust, prolonged depolarization driving complex spike burst (CSB) firing (*Bittner et al., 2015*). Such CSBs are thought to be driven by dendritic $Ca^{2+}$ spikes (plateau potentials) mediated by voltage-gated $Ca^{2+}$ channels and NMDA receptors (*Takahashi and Magee, 2009*; *Grienberger et al., 2014*). The formation of new PFs by plateau potentials is thought to be mediated by synaptic plasticity obeying a non-Hebbian rule, whereby the excitatory input synapses activated in a ~2-s-long time window around the CSB are rapidly potentiated. This form of plasticity was named behavioral time scale synaptic plasticity (BTSP; *Bittner et al., 2017*; *Milstein et al., 2021*). After initial electrophysiological demonstration, spontaneous PFF with BTSP-like characteristics was observed in CA1PCs by in vivo $[Ca^{2+}]$ imaging as well (*Grienberger and Magee, 2022*; *Priestley et al., 2022*; *O'Hare et al., 2025*). PFF with features similar to spontaneous BTSP-like PFF could also be induced artificially by strong sustained electrical or optogenetic depolarization of a cell (*Bittner et al., 2015*; *Bittner et al., 2017*; *Diamantaki et al., 2018*; *Milstein et al., 2021*; *Gonzalez et al., 2025*; *Rolotti et al., 2022*; *Fan et al., 2023*), a stimulus assumed to evoke plateau potentials. In summary, BTSP evoked by CSBs has been proposed to serve as the main mechanism driving de novo PFF; however, its triggering mechanisms are enigmatic and its prevalence under different behavioral conditions is not yet known.

Whereas BTSP is a suitable and attractive candidate, other mechanisms may also contribute to PFF. In vivo patch-clamp recordings (*Cohen et al., 2017*) from CA1PCs in head-fixed mice running in a novel virtual environment showed that most of the delayed PFFs did not begin with CSBs, but with a weak subthreshold spatially tuned depolarization that increased when the PF emerged. This result suggests the involvement of synaptic plasticity mechanisms other than BTSP, such as subthreshold cooperative plasticity mechanisms operating locally in dendrites (*Golding et al., 2002*; *Mehta, 2004*; *Dudman et al., 2007*; *Sjöström et al., 2008*; *Kim et al., 2015*; *Sheffield and Dombeck, 2019*; *Magó et al., 2020*) with the potential contribution of conventional Hebbian plasticity after initial spiking begins (*Magee and Johnston, 1997*; *Markram et al., 1997*; *Bi and Poo, 1998*). Furthermore, network mechanisms may also drive the emergence of new PFs. It has been shown that optogenetic stimulation of a small subset of CA1PCs at a particular location induced PFF or remapping in non-stimulated cells, suggesting that changes in synaptic coupling and circuit dynamics of PCs and inhibitory interneurons can also lead to reorganization of spatial maps by unmasking latent PFs (*McKenzie et al., 2021*). Also pointing to a network effect, optogenetic induction of BTSP-like PFF has been shown to be restricted by lateral inhibition, strongly limiting the number of CA1PCs that can simultaneously form new PFs by BTSP (*Rolotti et al., 2022*). Altogether, a comprehensive understanding of the contribution of possibly diverse cellular mechanisms underlying PFF during various forms of remapping processes is lacking.

BTSP has several unique features when probed by somatic $[Ca^{2+}]$ imaging (*Grienberger and Magee, 2022*; *Priestley et al., 2022*). First, the prolonged CSB results in large $[Ca^{2+}]$ transients during the initial PFF event, typically followed by weaker $Ca^{2+}$ signals on consecutive traversals through the PF. Second, due to the long and asymmetric temporal kernel of the plasticity (favoring potentiation of

inputs active 1–2 s before the CSB), a substantial backward shift in the spatial position of the PF center can be observed on linear tracks after the formation lap. Third, the width of the new PF is generally proportional to the running speed of the animal during the PFF event. In contrast, PFF by other synaptic or network mechanisms is not expected to exhibit such properties. Using these features, we aimed to characterize PFFs as mediated by BTSP- or non-BTSP-like mechanisms in large populations of CA1PCs. We performed two-photon [$Ca^{2+}$] imaging of CA1PCs in head-fixed mice performing a spatial navigation task in familiar and partially novel virtual corridors. We find that BTSP is prevalent but is not the only mechanism driving PFFs, as many new PFs formed without exhibiting any of the characteristics of BTSP. Furthermore, we show that robust CSB-like somatic activity is often ineffective in inducing new PFs, suggesting that additional factors are required than just CSBs to successfully initiate new PFs.

## Results

### Place modulated activity of CA1PCs in mice navigating in a virtual corridor

In vivo two-photon [$Ca^{2+}$] imaging of CA1PCs was performed from the dorsal hippocampus of head-restrained double transgenic mice (Thy1-GCaMP6s and Cre-dependent td-Tomato), injected with diluted Cre-recombinase expressing AAV (*Figure 1A, B*). Mice were trained to run in an ~8-m-long virtual corridor for a water reward. The wall of the virtual corridor contained six well-discernible visual landmarks, the last of which indicated the reward zone (RZ), where mice were required to lick for water reward (*Figure 1C*). Well-trained mice ran with a relatively constant speed of ~30–40 cm/s through the corridor and slowed down and started licking just before entering the RZ (*Figure 1D*), indicating that they learned the position of the RZ. Imaging was performed in well-trained mice across multiple days (*n* = 163 imaging sessions in 45 mice). The imaging field of view contained ~1000 CA1PC somata (*Figure 1B*), which showed highly variable activities during the session (*Figure 1E–H*); some cells were silent whereas others showed moderate to high activity rates. Plotting the activity as a function of the animal's location in the virtual corridor showed high variability in the spatial tuning of active cells (*Figure 1I*, for PF determination see Methods). The PFs of place cells tiled the entire length of the corridor with an increased density around the visual landmarks (*Figure 1J*). Place cells had a variable number of PFs with widely different widths, covering a variable fraction of the virtual corridor (*Figure 1K*). These results demonstrate that the activity of CA1PCs in our experimental conditions is comparable to that observed in animals exploring large environments (*Rich et al., 2014*; *Lee et al., 2020*; *Eliav et al., 2021*).

### New PFs are formed with both BTSP- and non-BTSP-like mechanisms

During navigation in the familiar virtual corridor, new PFs often appeared suddenly at various spatial locations during the session (*Figures 2 and 3*). To investigate the possible mechanisms underlying such a robust change of activity, we examined the dynamics of the newly formed PFs. If BTSP is the mechanism generating new PFs in CA1PCs, the PFs are expected to have the following characteristics: first, since BTSP is induced by a plateau potential and consequential prolonged CSB (*Bittner et al., 2015*; *Bittner et al., 2017*), we expect that the amplitude of the [$Ca^{2+}$] transient in the formation lap is larger than that in the following laps. Second, because of the asymmetrical temporal kernel of BTSP (*Bittner et al., 2017*), the center of mass (COM) of the new PF is expected to shift to the backward direction compared to the peak activity in the formation lap. To quantify these dynamics, we calculated two parameters for each newly formed PF: the formation lap gain and the initial COM shift (see Methods and *Priestley et al., 2022*). We then categorized all newly formed PFs in our recordings based on these two parameters. Using a conservative approach, we categorized a new PF to be formed by a BTSP-like mechanism if it had both positive gain and negative shift values (*Figure 2A*; *n* = 311 new PFs), whereas new PFs exhibiting neither positive gain nor negative shift were considered as non-BTSP-like events (*Figure 2B*; *n* = 58). All other newly formed PFs (no-gain with backward shift and gain without backward shift) were excluded from further analysis (see Methods for fractions of excluded PFs).

We considered the possibility that PFF events might be erroneously classified as non-BTSP-like due to a random occurrence of a $Ca^{2+}$ event in the future PF in the lap preceding a BTSP-like PFF. However,

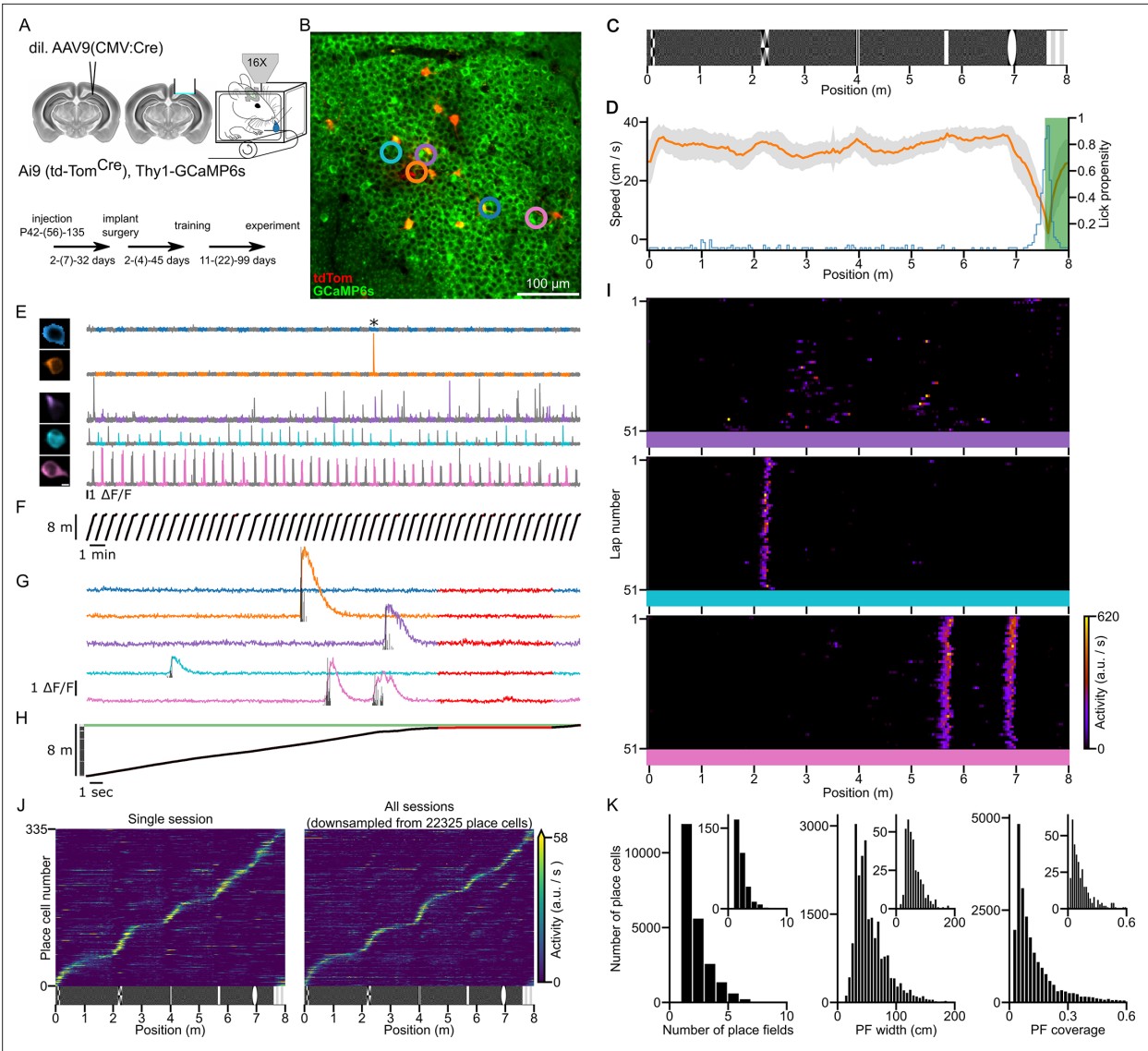

**Figure 1.** In vivo two-photon [Ca$^{2+}$] imaging from head-restrained mice during spatial navigation in a virtual corridor. (**A**) Double transgenic mice (GCaMP6s and Cre-dependent td-Tomato) were injected with a diluted AAV expressing Cre-recombinase for sparse td-Tomato labeling. Following craniotomy and cannula implantation above the left dorsal CA1 area, animals were trained and imaged with a two-photon (2P) microscope during navigation in an ~8-m-long virtual corridor. The timeline shows the minimum, (median), and maximum number of days between the procedures. (**B**) Mean 2P image of an imaging field of view. Example cells shown in panels (**E**), (**G**), and (**I**) are indicated by colored circles. (**C**) Wall pattern of the virtual corridor consisting of low contrast random checkerboard background with six high contrast visual landmarks. (**D**) Running speed (mean: orange; ± SD: gray; single session) decreased and lick propensity (blue) increased before the animal reached the reward zone (green area). (**E**) ROI masks (left; scale bar: 5 μm) and [Ca$^{2+}$] signals (right) of the cells marked in panel (**B**) during a single session. Gray segments correspond to odd laps in the virtual environment. (**F**) Position of the animal along the corridor aligned with the fluorescence activity shown in panel (**E**). (**G, H**) Same as panels (**E**) and (**F**) on an extended time scale showing a single lap (indicated by * in (**E**)). The time period when the animals' running speed was below 5 cm/s is shown in red and was excluded from the analysis. Black vertical bars are inferred activity. (**I**) Raster plots showing the inferred neuronal activity of three cells (color coded cells in panels (**B**), (**E**), and (**G**)) as a function of the lap number and spatial location of the animal. (**J**) Coverage of the virtual corridor by place fields (PFs). Tuning curves from cells with at least one significant PF are included. The tuning curves are ordered by the location of their largest peak activity. Left panel: Data from a single session. Right panel: Data from all experiments (22,325 place cells recorded across 163 sessions from 45 mice) was randomly down-sampled to match the sample size of the single session. (**K**) Distributions of the number of PFs of place cells, PF widths, and PF coverage (proportion of the length) of the virtual corridor. Data are presented for all sessions and for a single session (insets).

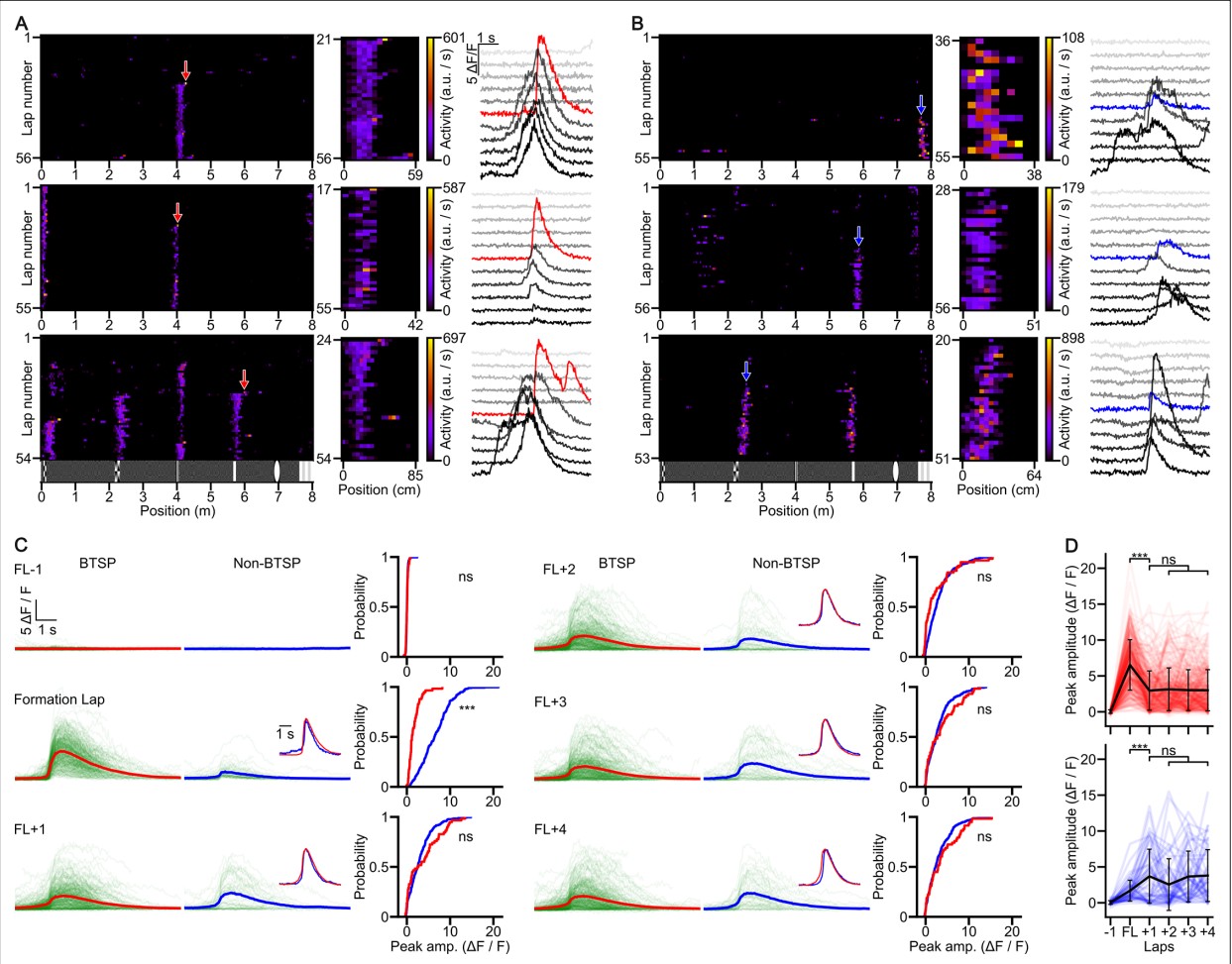

**Figure 2.** Comparison of [Ca²⁺] transients underlying different place field formation (PFF) events. (**A**) Behavioral time scale synaptic plasticity (BTSP)-like PFFs. Activity raster plots (left) show three cells with BTSP-like PFFs (red arrows). The middle panels zoom in on the activity of newly formed PFs marked by red arrows. Fluorescence traces (right) are shown during PF traversal before, during (red traces), and after PFF (±5 laps). (**B**) Same as (**A**), but for non-BTSP-like PFFs (blue arrows and traces). (**C**) Comparison of [Ca²⁺] transients before, during, and after PFF. Fluorescence traces for BTSP-like (left; mean trace in red) and non-BTSP-like (middle, mean trace in blue) PFFs are shown for each lap, from the lap before the formation lap (FL−1) to the fourth lap post-formation (FL+4). Traces were aligned to the midpoint of the steepest rising slope for detectable [Ca²⁺] transients, or to the time point of PF center traversal for traces lacking transients. Insets: Peak-normalized and peak-aligned mean traces are superimposed. Cumulative distributions of the peak [Ca²⁺] transient amplitudes for BTSP (red) and non-BTSP (blue) events across each lap are shown on the right. (**D**) Progression of peak [Ca²⁺] transient amplitudes (ΔF/F) during individual BTSP (red) and non-BTSP (blue) PFF events. Thin lines connect data points in consecutive laps for each PFF. Black circles with whiskers are lap-wise population means ± SD (BTSP: $n$ = 311 events, non-BTSP: $n$ = 58 events from 29 mice). Statistical analysis revealed that the amplitude of the [Ca²⁺] transients in the formation lap is significantly higher during BTSP-like compared to non-BTSP-like PFFs and the amplitudes in the other laps are not significantly different (mixed ANOVA: main effect for group (BTSP/non-BTSP): $p$ = 0.038; lap: $p < 0.001$, interaction: $p < 0.001$; Tukey post hoc test: formation lap: $p < 0.001$, all other laps: $p > 0.711$). *** $p < 0.001$.

The online version of this article includes the following figure supplement(s) for figure 2:

**Figure supplement 1.** Characteristics of place field formation (PFF) and stability.

**Figure supplement 2.** Place field formations (PFFs) in a VR corridor with a complex wall pattern.

a bootstrap analysis (see Methods) demonstrated that this is unlikely for the majority (88%) of non-BTSP-like new PFF events (*Figure 2—figure supplement 1A*).

Our standard VR environment is dominated by six robust visual landmarks, around which PFs (*Figure 1J*) and new PFF events (*Figure 3A*) are frequent. To investigate new PFF events in a visually more complex VR environment, we created a corridor with rich and complex wall texture without dominant visual landmarks (*Figure 2—figure supplement 2A*). Changes in running speed and lick rates just before entering the RZ at the end of the corridor (*Figure 2—figure supplement 2B*) indicated

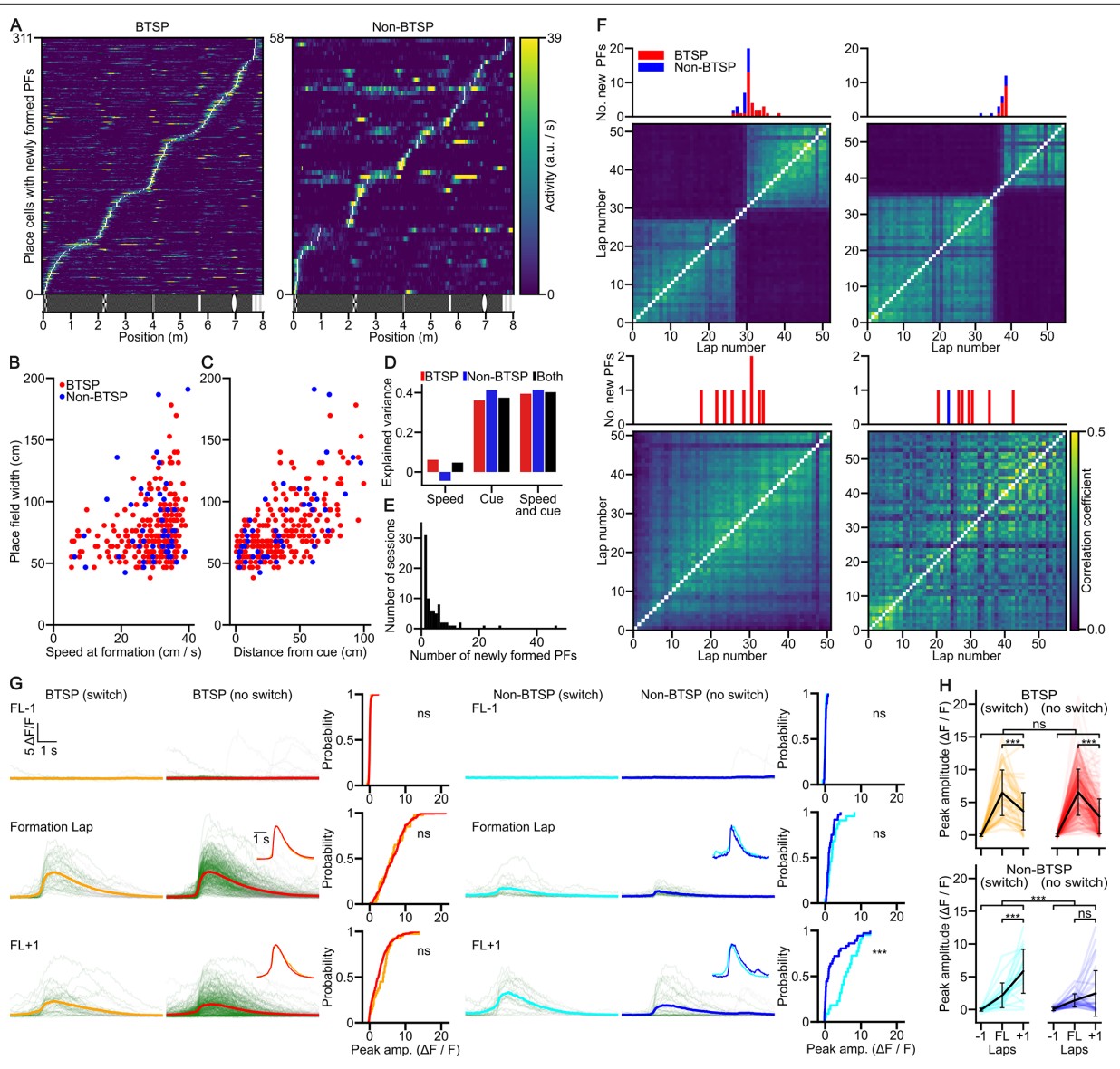

**Figure 3.** Newly formed place fields (PFs) cover the entire virtual corridor and have non-uniform birth rates. (**A**) The tuning curves of behavioral time scale synaptic plasticity (BTSP)- (left) and non-BTSP-like (right) PFs were ordered by the center position of the newly born PFs (white space bins). (**B**) The width of BTSP-like newly formed PFs is correlated with the animal's running speed during PFF. Scatter plot shows individual BTSP- (red) and non-BTSP-like (blue) PF formation (PFF) events (Spearman correlations: BTSP ($n$ = 261): $R$ = 0.29, p = 2.7 × 10$^{-6}$; non-BTSP ($n$ = 47): $R$ = 0.015, p = 0.92). (**C**) Correlation between the width of PFs formed by BTSP- (red) and non-BTSP-like (blue) mechanisms and the distance from the nearest visual landmark (Spearman correlations: BTSP ($n$ = 261): $R$ = 0.59, p = 1.4 × 10$^{-25}$; non-BTSP ($n$ = 47): $R$ = 0.67, p = 2.4 × 10$^{-7}$). (**D**) Explained variance of PF width by the running speed in the formation lap, distance to nearest visual landmark at formation, or both, for BTSP- (red) non-BTSP- like (blue) or combined (black) PFF events. (**E**) Histogram of the number of newly formed PFs per single session in the virtual corridor. A few sessions have unusually high (>20) new PFF events. Note that the analysis excludes PFFs in the first 17 laps of the session (Methods). (**F**) Number of newly formed PFs by laps. Example sessions showing high number of BTSP- (red) and non-BTSP-like (blue) PFFs (upper panels, two of the three sessions with the largest numbers of PFFs in the histogram in panel (**E**)) and sessions with moderate number of PFFs (lower panels). The histograms are aligned with lap-by-lap population vector (PV) correlograms, which reveal sudden change in population activity for the upper panels and lack of such representational switch for the lower panels. Color scale applies to all panels. (**G**) [Ca$^{2+}$] transients of PFF events during representational switches (switch, isolated from sessions with the three highest number of PFF events in the histogram in panel (**E**)) are compared to PFF events outside representational switches (no switch) for BTSP- (left) and non-BTSP-like (right) events. Traces are shown from the lap preceding PFF (FL−1) to the lap following PFF (FL+1). Traces were aligned to the midpoint of the steepest rising slope, or to the time point at which the PF center was reached (for traces without detectable transients). The mean traces are shown in orange, red, cyan, and blue. Insets: peak-normalized and peak-aligned mean traces. The cumulative distributions show the peak [Ca$^{2+}$] transient amplitudes for BTSP (with switch: yellow, without switch: red) and non-BTSP (with switch: cyan, without switch: blue) PFF events across

*Figure 3 continued on next page*

*Figure 3 continued*

each lap. (**H**) Progression of peak amplitudes ($\Delta F/F$) during individual switch and no switch BTSP (yellow, red) and non-BTSP (cyan, blue) formation events. Individual data points are connected to show the change in peak amplitude across laps for each event. Black circles with whiskers are lap-wise population mean ± SD (BTSP no switch: $n = 271$, BTSP switch: $n = 40$, non-BTSP no switch: $n = 36$, non-BTSP switch: $n = 22$ events from 29 mice). The peak amplitudes of [Ca$^{2+}$] transients are not significantly different for the BTSP-like events (mixed ANOVA: main effect for group (switch/no switch): p = 0.457; lap: p < 0.001, interaction: p = 0.214) and for the formation lap of the non-BTSP-like events but are significantly higher for non-BTSP-like events during switches than outside switches in the lap after the formation lap (mixed ANOVA: main effect for group (switch/no switch): p < 0.001; lap: p < 0.001, interaction: p < 0.001; Tukey's *post hoc* test: formation lap –1: $P=0.999$, formation lap: $P=0.774$, formation lap +1: $P<0.001$). *** p < 0.001.

The online version of this article includes the following figure supplement(s) for figure 3:

**Figure supplement 1.** Factors contributing to place field width and correlations between fluorescence and inferred spike rates.

**Figure supplement 2.** Spontaneous representation switches within a virtual environment.

that animals could also learn the location of the RZ in this complex VR environment. PFs of variable widths covered the entire VR corridor in a relatively uniform manner (*Figure 1J* and *Figure 2—figure supplement 2C, D*). Here, BTSP- (*Figure 2—figure supplement 2E*, $n = 117$ event from 12 imaging sessions from 4 mice) and non-BTSP-like (*Figure 2—figure supplement 2F*, $n = 18$) PFF events were readily observed with a very similar proportion (*Figure 2—figure supplement 2G*) to that observed in our original VR environment.

In summary, our analysis demonstrated that, although a large fraction of newly formed PFs was generated by a BTSP-like mechanism, a substantial fraction (~15%) of new PFF was produced by a different process.

Comparison of the [Ca$^{2+}$] transients of BTSP-like ($n = 311$) and non-BTSP-like ($n = 58$) new PFs in the laps before, during, and after their formation showed that, apart from the strongly different [Ca$^{2+}$] amplitudes in the formation lap, [Ca$^{2+}$] transients within the PF on the subsequent laps had similar amplitudes (*Figure 2C, D*), fractions of active laps (*Figure 2—figure supplement 1B*), widths (BTSP: 46.4 ± 24.4 cm; non-BTSP: 50.4 ± 32.5 cm, p = 0.28), peak rates (BTSP: 19.0 ± 14.7 a.u./s; non-BTSP: 21.4 ± 16.8 a.u./s, p = 0.27) and out-of-field firing rates (BTSP: 0.64 ± 0.68 a.u./s; non-BTSP: 0.83 ± 1.25 a.u./s, p = 0.09, all unpaired *t*-test), indicating that the basic characteristics of activity in the stabilized new PFs were comparable between the two groups. Furthermore, both types of new PFs exhibited similar distributions along the virtual track, tiling the whole corridor with increased densities around the visual landmarks (*Figure 3A*). The prevalence of PFF did not appear to be related to the time from first training session or the number of previous sessions (*Figure 2—figure supplement 1C*).

A further characteristic feature of BTSP is that, due to its seconds-wide temporal kernel, the width of the new PF is proportional to the running speed of the animal during the formation lap, that is the potentiated inputs cover a wider spatial sector if the animal travels longer distance during the plasticity time window (*Bittner et al., 2017*; *Priestley et al., 2022*). As an independent confirmation that newly formed PFs in our categorization of PF groups genuinely differed in the mechanisms underlying their formation, we found a significant positive correlation between formation lap running speed and PF width in the BTSP-like but not in the non-BTSP-like PF category (*Figure 3B*). Because of the difference in the number of events in the two groups, we randomly subsampled the BTSP-like events to the sample size of the non-BTSP-like PFF events 10,000 times and performed regression analysis. This bootstrapping revealed that both the *r* and p values of the fit to the non-BTSP data fell outside the 95% confidence interval of the bootstrapped BTSP values, indicating that the difference between the two groups was robust.

Because the running speed did not appear to fully account for the observed variance in PF width, we examined the contribution of our VR wall structure to PF width. Our standard VR wall pattern contained several narrow, salient visual landmarks, separated by longer corridor segments with a gray low-contrast pattern. We found that the animal's distance to the closest visual landmark center significantly correlated with PF width for both BTSP- and non-BTSP-like PFFs (*Figure 3C*, *Figure 3—figure supplement 1A, B*). Considering the effects of both speed and cue distance, we found that cue distance explained a similarly large fraction of the PF width variance for both BTSP- and non-BTSP-like PFFs, whereas speed explained a moderate proportion of PF width variance only in the case of BTSP-like PFFs (*Figure 3D*). These results demonstrate that in our VR environment, the complexity of the wall pattern is a major contributor to the PF width in both BTSP- and non-BTSP-like PFF events.

## Concentrated PFF associated with representational switch

We next examined the distribution of new PFF events across imaging sessions and laps. While in most of the sessions, the number of newly formed PFs was moderate, a few sessions contained unusually large numbers of such events (*Figure 3E*). When we examined the distribution of new PFFs in these sessions across individual laps, we found that the majority of PFF events occurred within a short period over a limited number (3–10) of consecutive laps (*Figure 3F*). These concentrated PFF periods were termed PFF 'surges', which included PFs formed both by BTSP- and non-BTSP-like mechanisms (non-BTSP: 22 events; BTSP: 40 events, non-BTSP/BTSP ratio: 0.55). The unusually high rate of new PFF events during PFF surges led us to investigate whether they were associated with a more general network-wide reorganization of the CA1PC representation of the virtual corridor. Indeed, we found that the lap-by-lap correlation of activity in the whole population of imaged CA1PCs underwent a drastic and rapid switch between two stable states before and after the PFF surge (*Figure 3F*, *Figure 3—figure supplement 2*). A closer look indicated that the initial representation began to disorganize as the surge started to ramp up and a new coherent representation started at the lap with the peak PFF rate. No such representational switches were observed in sessions without PFF surges (*Figure 3F*, bottom). The remapping was essentially global, as only a small fraction (5.4%) of CA1PCs retained their pre-switch tuning curve after the switch. When we compared the [Ca$^{2+}$] dynamics of PFFs during surges with those of PFFs outside the surges, we found similar [Ca$^{2+}$] amplitudes in both the BTSP- and non-BTSP-like groups, except for larger [Ca$^{2+}$] signals in the first lap after the formation lap in the non-BTSP group during surges compared to outside surges (*Figure 3G, H*). The period including the PFF surge and representational switch was not accompanied by any noticeable change in the behavioral performance of the animals as assessed by running speed and lick rate (*Figure 3—figure supplement 2A–C*). Furthermore, multiday imaging revealed that in some animals, switching between the same two representations occurred on consecutive days (*Figure 3—figure supplement 2D*), and occasionally, multiple representation switches within one imaging session could be observed (*Figure 3—figure supplement 2E*). Altogether, these results reveal unique, short epochs during task-engaged behavior when rapid reorganization of the population activity of CA1PCs takes place, in part due to new PFFs by both BTSP- and non-BTSP-like mechanisms.

## New PFF in a novel environment is mediated by both BTSP- and non-BTSP-like mechanisms

CA1PCs are known to rapidly remap in novel environments to create a specific representation of newly explored locations. We next sought to determine the contribution of BTSP- and non-BTSP-like mechanisms to new PFFs in a novel environment by extending the familiar virtual corridor with a 6-m-long segment decorated with never-seen visual landmarks during the imaging session (*Figure 4A*). Animals traversed this novel segment with an average speed comparable to that in the familiar portion of the maze. The maze extension did not induce a detectable change in lick frequency. At the end of the familiar segment of the 14-m-long corridor, the animals reduced their running speed and displayed anticipatory licking just before entering the RZ, similarly to their performance in the standard 8-m-long corridor. Population-level analysis revealed a reorganization of the neuronal representation, as demonstrated by a decreased population vector (PV) correlation between the extended and original maze, despite high intra-segment correlations (mean correlation coefficients: original maze odd vs. even: 0.50 ± 0.19, original maze odd vs. extended maze even: 0.08 ± 0.10, extended maze odd vs. even: 0.36 ± 0.20; n = 6 mice; repeated measures ANOVA (p < 0.001) followed by Tukey post hoc test: original maze odd − even vs. original maze odd − extended maze even: p < 0.001, extended maze odd − even vs. original maze odd − extended maze even: p = 0.002, original maze odd − even vs. extended maze odd − even: p = 0.114). Consistent with previous studies (*Sheffield et al., 2017*; *Dong et al., 2021*; *Priestley et al., 2022*), we observed many new PFs already during the first traversal in the novel extension of the virtual environment, and additional new PFs formed during subsequent laps, although with lower probability (*Figure 4A, B*). The frequency of new PFFs was the highest in the first few laps for both BTSP- and non-BTSP-like events (*Figure 4B*). Analysis of the somatic [Ca$^{2+}$] transients revealed that their amplitudes were very similar to those observed in the familiar environment for both the BTSP- and non-BTSP-like PFF events during the formation lap (*Figure 4C*). Interestingly, the relative prevalence of non-BTSP-like PFF was higher in the first laps of the novel environment than that observed in the familiar environments (novel corridor laps 1 and 2: n = 26 non-BTSP, n = 43 BTSP

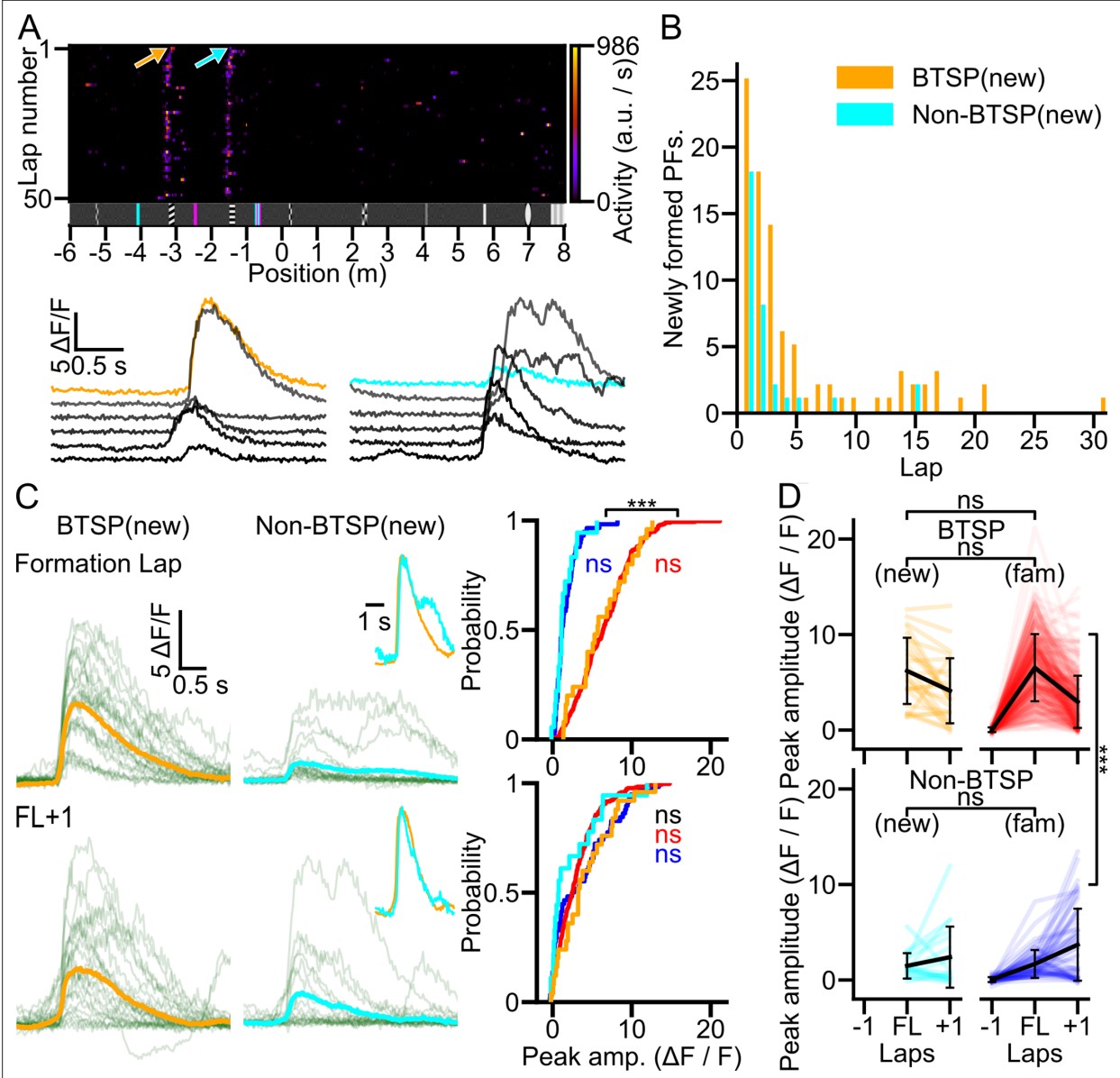

**Figure 4.** Similar [Ca²⁺] transients underlie place field formation (PFF) in familiar and novel environments. (**A**) PFF in a familiar environment with a novel extension. A familiar corridor was extended with a new 6-m-long segment. New PFs appear already during the first traversal (top) by both behavioral time scale synaptic plasticity (BTSP)- (red arrow) and non-BTSP-like (blue arrow) dynamics. Bottom, [Ca²⁺] transients during and after the PFFs marked on the raster plot. (**B**) Histogram of BTSP- and non-BTSP-like PFFs during the first session in the extended maze region ($n = 6$ mice). (**C**) [Ca²⁺] transients of the formation lap and the following lap (FL+1) of BTSP- (left panels, mean trace in orange) and non-BTSP-like (middle panels, mean trace in cyan) PFF events in the new extension. Insets show peak-normalized, peak-aligned mean traces. Cumulative distributions of the peak [Ca²⁺] transient amplitudes for BTSP and non-BTSP events across each lap are shown on the right. (BTSP$_{NEW}$, orange, $n = 24$, non-BTSP$_{NEW}$: cyan, $n = 16$ events) of the novel environment ($n = 6$ animals), and familiar environment (BTSP: red, non-BTSP: blue; data from *Figure 2*). (**D**) Progression of peak [Ca²⁺] transient amplitudes during BTSP (top) and non-BTSP (bottom) PFF events in the extended maze (new). BTSP and non-BTSP data from *Figure 2* (fam) are also shown. Thin lines connect data points in consecutive laps for each PFF. The amplitudes of [Ca²⁺] transients in the formation lap and the lap after formation were not significantly different between the new and familiar environments for both BTSP- and non-BTSP-like events (two-way mixed ANOVA: main effect for PFF type (BTSP/non-BTSP): $p < 0.001$, corridor (familiar/new): $p = 0.701$, type vs. corridor: $p = 0.193$, lap: $p = 0.018$, lap vs. type: $p < 0.001$, lap vs. corridor: $p = 0.738$, lap vs. type vs. corridor: $p = 0.025$; Tukey post hoc tests: lap 0: BTSP new vs. familiar: $p = 0.999$, non-BTSP new vs. familiar: $p = 0.999$; lap 1: BTSP new vs. familiar: $p = 0.633$, non-BTSP new vs. familiar: $p = 0.780$). *** $p < 0.001$.

in 6 mice, non-BTSP/BTSP ratio = 0.60; familiar corridor: $n$ = 58 non-BTSP, $n$ = 311 BTSP in 29 mice, non-BTSP/BTSP ratio = 0.19; Chi-square test: p < 0.001), whereas in the subsequent laps BTSP-like PFF dominated again (novel corridor region, laps 3–31: non-BTSP/BTSP ratio = 0.15).

## Large somatic [Ca²⁺] transients per se are not sufficient to induce new PFs

The above analysis demonstrated an important role for BTSP in new PFFs; however, it also indicated that BTSP is not an exclusive mechanism of PFF, as new PFs could also develop without strong initial somatic activity during the formation lap. Finally, we asked the following question: Is the occurrence of strong somatic Ca²⁺ activity sufficient to trigger the formation of new PFs? We selected CA1PCs exhibiting at least one BTSP-like newly formed PF in a given imaging session in the familiar corridor and identified all somatic Ca²⁺ transients in that session that had amplitudes larger than the [Ca²⁺] transient initiating the BTSP-like PFF (if multiple PFs were formed then the largest formation lap [Ca²⁺] transient was used; *Figure 5*). We found that, in approximately one quarter of the sessions, BTSP-inducing [Ca²⁺] transients could be matched with at least one larger [Ca²⁺] transient that failed to initiate new PFF ('solitary events', *Figure 5A, B*). Our results altogether suggest that large somatic [Ca²⁺] transients are neither indispensable nor are per se always sufficient to induce the formation of new PFs.

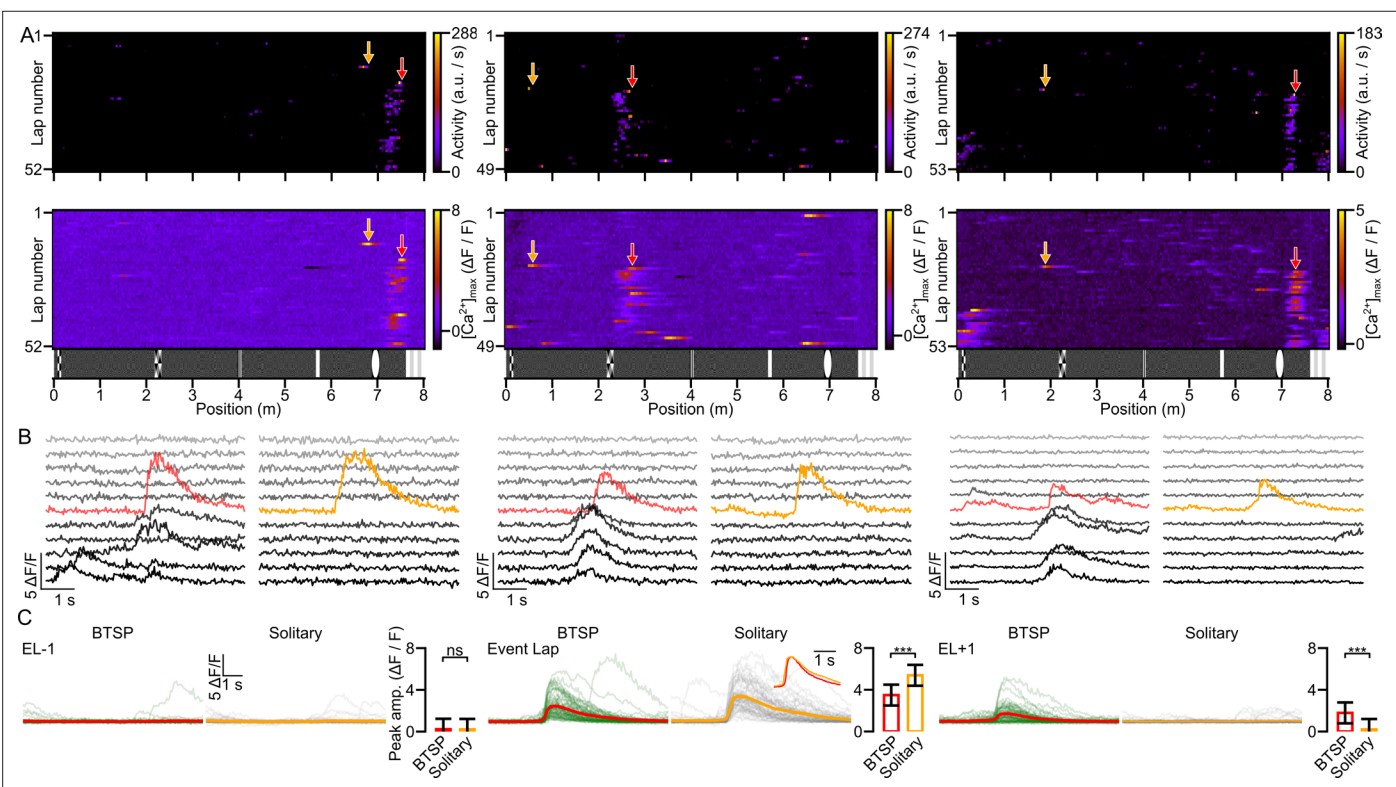

**Figure 5.** Large somatic [Ca²⁺] events are often not sufficient to evoke novel place fields. (**A**) Raster plots of inferred activity (top) and [Ca²⁺] transients (bottom) vs. spatial location in each lap for three cells reveals behavioral time scale synaptic plasticity (BTSP)-like new place field formations (PFFs; red arrows) as well as large amplitude activities (yellow arrows) which did not evoke new PFs (solitary events). (**B**) Fluorescence traces around the BTSP-like PFF events (red) marked in panel (**A**) and larger amplitude [Ca²⁺] transients not associated with PFF (yellow) within the same sessions are shown. (**C**) Traces from the lap before (EL−1), during (Event Lap), and the lap after (EL+1) of BTSP-like PFF events (left traces) and solitary non-PF-evoking events (right traces) identified within the same session. In EL−1, traces were aligned to the time point of crossing the center of the PF. Pre- and post-solitary event traces were aligned to the crossing time of the first space bin of the corresponding solitary event. Inset: peak-normalized and peak-aligned mean fluorescence traces. Bar graphs depict the peak amplitudes of [Ca²⁺] transients (mean ± SD; BTSP: $n$ = 59, solitary: $n$ = 59 events, $n$ = 22 mice) in the given laps. The amplitudes of [Ca²⁺] transients of the solitary events in the event lap were significantly higher, whereas in the EL+1 they were smaller compared to those of BTSP-like events (mixed ANOVA: main effect for group (BTSP/solitary): p = 0.682, lap: p < 0.001, interaction: p < 0.001; Tukey post hoc test: lap −1: p = 1, event lap: p < 0.001, lap +1: p < 0.001). *** p < 0.001.

## Discussion

In the present study, we performed in vivo two-photon [$Ca^{2+}$] imaging from hundreds of dorsal hippocampal CA1PCs while mice ran for a water reward in familiar and partially novel virtual environments. During navigation in familiar and novel corridors, PFF events occurred in CA1PCs spontaneously along the entire corridor. Many of these events had characteristics reminiscent of those reported earlier for BTSP-like PFF events, but a significant fraction of PPFs had small [$Ca^{2+}$] transient amplitudes in the formation lap and did not show the characteristic backward shift in the COM of the newly formed PF. Such non-BTSP-like PFFs not only occurred spontaneously in familiar environments, but also during representational switches within familiar and in the first laps of the exploration of a novel environment. We also observed that in many sessions when a CA1PC displayed a BTSP-like PFF event, there were other large [$Ca^{2+}$] transients that did not induce novel PFs (solitary events) even though they had higher peak amplitudes than that of the BTSP-like PFF inducing [$Ca^{2+}$] transient. These results demonstrate that BTSP-like PFF is an important but not exclusive mechanism of new PFFs and that large somatic [$Ca^{2+}$] transients are not always sufficient per se to induce new PFs. We note, however, that our approach using somatic [$Ca^{2+}$] imaging does not provide information on dendritic [$Ca^{2+}$] signals; therefore, we cannot be sure that the large-amplitude somatic events indicate dendritic plateau potentials. In addition, the indirect relationship between electrical activity and $Ca^{2+}$ signals and the likely limited detectability of single APs under our experimental conditions also adds some uncertainty to our interpretations.

The lack of large initial somatic [$Ca^{2+}$] transients – that are likely associated with strong bursts involving $Ca^{2+}$ plateaus – during non-BTSP-like PFFs suggests two possible scenarios: they either do not involve excitatory synaptic plasticity in CA1PCs but rather result from a change in incoming excitatory or inhibitory synaptic inputs, or they are driven by other synaptic plasticity mechanisms that do not require global $Ca^{2+}$ plateaus and CSBs to increase excitatory synaptic weights. Local synaptic plasticity in individual dendritic branches is a particularly intriguing possibility for the latter scenario, since it has been shown that PFF can be preceded by local dendritic activity at the future PF locations (*Sheffield et al., 2017*). This activity, presumably produced by clustered activation of Schaffer collateral synapses (*Adoff et al., 2021*), may induce local synaptic potentiation, whereby the inputs eventually become suprathreshold at the soma manifesting as non-BTSP-like PFF. Importantly, long-term potentiation (LTP) of excitatory synaptic inputs by local dendritic $Na^+$ and/or NMDA spikes (d-spikes) can be induced by a single or a few repetitions of d-spikes (*Remy and Spruston, 2007*; *Magó et al., 2020*), and thus can be consistent with rapid generation of new PFs. It remains to be explored whether the temporal coincidence kernel of local d-spike-induced LTP is on milliseconds or seconds time scale, that is different from or similar to that of BTSP. In contrast to synaptic plasticity induced by d-spikes, LTP induced by cooperative activity of clustered excitatory inputs without evoking APs or d-spikes (*Magó et al., 2020*) or LTP induced by conventional forms of Hebbian plasticity, for example spike timing-dependent plasticity (STDP, *Magee and Johnston, 1997*; *Markram et al., 1997*; *Bi and Poo, 1998*) requires larger number of repeated co-activations of pre- and postsynaptic cells and thus may be incompatible with the fast dynamics of PFF. Nevertheless, it cannot be ruled out that non-BTSP-like PFFs may involve STDP once the cell starts to fire APs, as it has been proposed to contribute to changes in rate, shape, and position of PFs (*Mehta et al., 1997*; *Mehta, 2015*, but see *Madar et al., 2025*). Since STDP in CA1PCs has been characterized under in vitro conditions, it is yet unknown whether its induction rules are similar in vivo, especially in the presence of active neuromodulatory systems (e.g. *Fisher et al., 2017*). Furthermore, BTSP may also reinforce already existing weak PFs (*Sheffield and Dombeck, 2015*; *Vaidya et al., 2025*).

By showing that both BTSP- and non-BTSP-like PFFs can underlie changes in spatial tuning of CA1PCs and by dissecting their relative contribution under different remapping conditions, our study can reconcile diverse and sometimes contradictory earlier observations. Although BTSP-like PFF events have been suggested recently as a dominant mechanism of new PFFs in the hippocampus (*Bittner et al., 2015*; *Bittner et al., 2017*; *Milstein et al., 2021*; *Grienberger and Magee, 2022*; *Rolotti et al., 2022*), previous results also indicated that PFs in a new environment frequently form without CSBs (*Cohen et al., 2017*). Recording from a large number of CA1PCs, we could estimate the prevalence of BTSP- and non-BTSP-like PFFs during various forms of remapping. We found that BTSP dominates PFFs under conditions when new PFs are born in individual CA1PCs that are part of a relatively stable representation in a familiar environment, i.e. during gradual representational

drift. In contrast, the relative ratio of non-BTSP-like PFF was much higher in the first laps of a novel corridor segment. This may be consistent with a global remapping process in response to reconfiguration of the excitatory input patterns arriving to individual CA1PCs from upstream areas. As a result, in some CA1PCs, no plasticity would be required to immediately generate a new PF, while in other CA1PCs, plasticity of the newly activated inputs (e.g. by activating local d-spikes) may also contribute to non-BTSP-like PFF, with this process possibly facilitated by a change in brain state due to the novelty of the new corridor segment. These processes altogether could increase the contribution of non-BTSP-like PFF to the initial development of the novel representation of the newly explored corridor.

Our results are consistent with previous reports of increased PFF during initial exposure to novel environments (*Priestley et al., 2022*) and around newly introduced salient cues in a familiar environment (*Grienberger and Magee, 2022*), although certain differences may be apparent. In particular, *Priestley et al., 2022*, who classified BTSPs similarly to our method but pooled all PFF events not clearly BTSP as an 'other' group, observed increased fraction of BTSP-like PFF on the first trial in a novel environment compared to a familiar environment (*Priestley et al., 2022*). Here, however, we have found an increase in the prevalence of non-BTSP-like PFF events during the initial exposure to the novel corridor. Yet, these results are not directly comparable for the following reasons: first, we defined the non-BTSP-like group as a subclass of 'other' PFFs based on multiple feature criteria to clearly separate them from BTSP-like PFFs; second, in the familiar environment, we only analyzed PFs that were formed after a silent period of at least 17 laps, whereas *Priestley et al., 2022* analyzed all PFs including those with established activity from the first lap in an environment.

Interestingly, we also observed spontaneous surges of coordinated new PFF events during short imaging epochs involving a few laps in the familiar corridor. These PFF surges included a high fraction of both non-BTSP- and BTSP-like PFF events and were associated with global remapping of the CA1 representation. The precise trigger and relevance of these representational 'switches' in a well-learned, invariant environment is currently unknown, but we speculate that – in the absence of alterations of sensory cues and task demands – they may indicate a change in the internal state of the animal, or a transition between relying on different signals for navigation (e.g. sensory cues vs. self-motion), assigning distinct internal contexts to the same environment (*Sanders et al., 2020*). Although different CA1 population codes across consecutive (interrupted) explorations of the same maze have been demonstrated (*Sheintuch et al., 2020*), to our knowledge, similar representational switches during a continuous behavioral epoch in an unchanged environment have only been described in the entorhinal cortex (EC; *Low et al., 2021*). It is possible that the representation switches we observed originate in the EC, which thus provides globally remapped inputs to the hippocampus similarly to the situation in a novel corridor. Overall, the higher ratio of non-BTSP- to BTSP-like PFFs in the initial laps of the novel environment and during representational switches (both ~0.60) than that in the familiar environment (0.19) suggests that distinct cellular/synaptic mechanisms might contribute to gradual representational drifts and abrupt global remapping.

While sudden large $Ca^{2+}$ activity at a location where a cell was previously inactive could often induce BTSP-like PFF, our results also demonstrate that a large fraction of even stronger $Ca^{2+}$ activities failed to initiate PFF in the same cell. If we assume that the large (and often prolonged) somatic $Ca^{2+}$ signals represent CSBs, this indicates either that the synaptic plasticity induced by the underlying plateau was not sufficiently strong to lead to somatic firing (perhaps influenced by head-fixed VR navigation with diminished sensory cues, *Ravassard et al., 2013*), or that the induction of plasticity may be gated by additional factor(s), for example modulatory signals or inhibition. This latter possibility highlights the importance of elucidating the molecular components contributing to the initiation, expression, and regulatory mechanisms governing the long-term increase in synaptic strength by BTSP. While research efforts of the past three to four decades provided an enormous amount of information regarding the molecular mechanisms underlying Hebbian long-term synaptic plasticity, the molecular events taking place during BTSP are almost completely unknown. Further research will be crucial to reveal the similarities and differences in the molecular mechanisms underlying these fundamentally different synaptic plasticities, which will allow us to design molecular modifications that specifically alter only one of them to uncover its consequences on new PFFs and animal behaviors.

# Materials and methods

## Animals

A total of 45 (23 male and 22 female) Ai9xGP4.3 double-transgenic mice were used in this study. The mice were generated by crossing two lines from The Jackson Laboratory: the Ai9 Cre reporter strain (IMSR_JAX:007909) conditionally expressing tdTomato and the GP4.3 line (IMSR_JAX:024275), which expresses the GCaMP6s calcium indicator in excitatory neurons under the *Thy1* promoter. Mice were housed in the vivarium of the HUN-REN Institute of Experimental Medicine under a standard 12-hr light/dark cycle with ad libitum access to water and food. Water intake was restricted during the experiments (details provided below). All procedures adhered to the regulations outlined in the Hungarian Act of Animal Care and Experimentation (40/2013 [II.14]) and received approval from the Animal Committee of the Institute of Experimental Medicine, Budapest (PE/EA/00874-5/202).

## Virus injection

Mice were first anesthetized with an intraperitoneal injection of a mixture of ketamine hydrochloride (CP-ketamine, 62.5 mg/kg), xylazine (CP-xylazine, 12.5 mg/kg), and promethazine hydrochloride (Pipolphen, 6.25 mg/kg). Local anesthesia was achieved by subcutaneous injection of ropivacaine hydrochloride (Naropine, 50–100 µl, 2 mg/ml). To protect the eyes, Opticorn A ointment was applied. The animal's head was then fixed in a stereotaxic frame, and a craniotomy of ~0.5 mm in diameter was made above the injection site. Four times 25 nl AAV vector (pENN.AAV.hSyn.Cre.WPRE.hGH, $3.3 \times 10^{10}$ GC/ml, Addgene #105553-AAV9) was injected into the left hemisphere at coordinates AP: –2, DV: –1.1, ML: 1.8, with 2.3 nl/s bouts separated by 1-min intervals. Following surgery, buprenorphine hydrochloride (Bupaq, 0.1 mg/kg) and carprofen (Rimadyl, 25 mg/kg) were administered subcutaneously for analgesia and preventing inflammation. Carprofen (35 ng/ml) was also supplemented in the drinking water for the following 2 days of postoperative care.

## Implant surgery

After virus injection (median: 7, range: 2–32 days) animals were anesthetized with Isoflurane (1.5% in carbogen, flow rate 0.8 l/min). A 3-mm diameter craniotomy was created over the virus injection site and the cortex above the hippocampus was gently aspirated. A 3-mm diameter 1.7-mm long stainless-steel cannula with a glass coverslip attached to the bottom was inserted and glued to the surrounding skull. A custom-made 3D printed titanium head bar was cemented to the skull together with the cannula. Buprenorphine hydrochloride (Bupaq, 0.1 mg/kg) and carprofen (Rimadyl, 25 mg/kg) were administered subcutaneously for analgesia and preventing inflammation. During the 2 days of postoperative care, the animal's drinking water was supplemented with 35 ng/ml of carprofen.

## Behavioral training

Water restriction started 24 hr before behavioral training (median: 4, range: 2–45 days after surgery). Throughout training days, animals received 5% sucrose solution as a reward during behavioral sessions. If an animal's weight fell below 85% of its starting weight by the end of the training/imaging session, the animal was provided free access to water for 10 min in its home cage. On the first day of training, animals were habituated to head-fixation in the behavioral setup and were encouraged to run on a virtual reality attached treadmill by receiving rewards via a lick-port at regular intervals. A virtual corridor was displayed on three flat screens placed in front and on the two sides of the animal's head and synchronized with the movement of the treadmill (PyOgre virtual reality and Luigs & Neumann treadmill logged and synchronized via a LabVIEW software). The animal was also rewarded for licking the lick-port in the designated RZ. Upon reaching the end of the corridor, animals were instantly teleported to the start. On the following days, the distance between running-related rewards was gradually increased until animals relied solely on operant rewards in the RZ at the end of the corridor. Subsequently, the length of the virtual environment was gradually extended from 2 to ~8 m by adding new segments to the beginning of the corridor. Animals were trained until they completed at least 50 laps within 30 min (median = 22, range 11–99 days, *Figure 2—figure supplement 1C*).

## In vivo two-photon [Ca²⁺] imaging

Two-photon [Ca$^{2+}$] imaging of CA1PCs was performed in vivo in the CA1 region of the left dorsal hippocampus in well-trained animals running typically 50 laps per session. A Femto-2D dual two-photon microscope equipped with a Nikon LWD 16X objective lens (NA = 0.80) was used, and the laser power was set to ~70 mW at the specimen, with a wavelength of 920 nm (Chameleon Vision, Coherent). Images of 512 by 505 pixels were acquired at a frame rate of ~31 Hz with a spatial resolution of 0.89 × 0.89 μm/pixel.

## Analysis of in vivo [Ca²⁺] imaging data

Motion correction, ROI detection, fluorescent trace extraction, and neuronal activity deconvolution were performed using the Suite2p software *Pachitariu et al., 2017* followed by analysis with custom-written Python scripts.

To account for background fluorescence originating from neuropil (tissue consisting of neuronal and glial processes surrounding the cell bodies), we performed neuropil correction. The corrected fluorescence ($F$) for each ROI was calculated by subtracting a scaled version (0.7x) of the Suite2p-estimated neuropil fluorescence ($F_{neu}$) from the raw fluorescence values ($F_{raw}$): $F = F_{raw} - 0.7 \times F_{neu}$. For a conservative estimation of inferred neuronal activities, we used the Suite2p spks output containing neuronal activity events deconvolved from the underlying fluorescence traces. We filtered the spks output by the following process. Local noise was assessed around each non-zero spks value. Local maxima of the neuropil-subtracted fluorescence ($F$) were identified between each non-zero spks value and the following ninth frame. These maxima were replaced with the average value of three neighboring fluorescence data points around the peak ($F_{max}$). Next, a linear baseline was fit to the fluorescence data points from the −30th to the −20th frame preceding each non-zero spks value. The local noise of the fluorescence ($SD_{local}$) was calculated as the root-mean-square error of the linear fit. The baseline fluorescence ($F_{base}$) for each activity was defined as the linear prediction at the −20th frame position. Global noise was characterized by the standard deviation ($SD_{global}$) of $F$ values within the range 1.5 times below and 0.5 times above the median $F$ value of the entire fluorescence trace. A non-zero spks value was considered valid only if the corresponding fluorescence transient amplitude ($F_{max} - F_{base}$) exceeded both the local and global noise. The resulting noise-reduced inferred activity values were used for all subsequent analyses.

To measure the amplitude of [Ca$^{2+}$] transients in laps before, during, and after PFFs, we calculated the change in fluorescence relative to baseline ($\Delta F/F$), according to the equation: $F - F_0/F_0$, where $F$ is the neuropil-corrected fluorescence and $F_0$ is the rolling median of $F$ derived from the preceding 1000 fluorescence values. For events earlier than the 1000th time point, we used the following 1000 time points for median calculation. The $\Delta F/F$ traces from different ROIs during PF transitions were aligned using the following procedure. In case at least one inferred spike occurred during the transition, the $\Delta F/F$ traces were smoothed using a five-point wide symmetrical moving average, and the difference between each smoothed point and the fourth point before it was calculated. The position of the maximum in this difference curve, corresponding to the steepest rising phase of the fluorescence transient, was used for trace alignment. For PF transitions without inferred events, the traces were aligned to their midpoint. Following trace alignment, an asymmetrical window encompassing −50–150 data points around the alignment point was isolated. The mean $\Delta F/F$ trace within this window across all transitions was calculated, and the position of its maximum determined. The peak amplitude for each trace was then calculated by subtracting the median $\Delta F/F$ value of the isolated segment from the $\Delta F/F$ value at the maximum of the mean trace. Finally, average and standard deviation of these peak amplitudes were calculated across all traces.

## Newly formed PF demarcation and categorization

### Spatial demarcation

The 832-cm long virtual environment was discretized into 4.24-cm wide space bins and data from bins with average running speed lower than 4 cm/s were excluded. Odd and even laps were separated and used for calculating mean inferred activity in each spatial bin across laps (odd and even tuning curves). ROIs with smaller than 0.25 Pearson's product-moment correlation coefficient between odd and even tuning curves were excluded from further analysis. We assessed spatial information in each ROI using Skaggs's metric (*Skaggs et al., 1992*). Statistical significance was assessed using a bootstrapping

procedure performed in Python. The inferred activity data for each lap was circularly shifted with a random amount in time, and Skaggs's information content was recalculated 1500 times to create a null distribution. ROIs with Skaggs's score exceeding the 99th percentile of the shuffled distribution were selected for further processing (*Eliav et al., 2021*). Tuning curves for each ROI were calculated as the mean inferred activity in each spatial bin across laps. Peaks and their footprints, the area they cover on the space axis, in the tuning curves were identified using functions from the SciPy library: find_peaks (peak_height = 6), peak_prominences, and peak_widths (rel_height = 0.93). Adjacent peaks were merged if their footprints overlapped or the intervening minimum value between peaks was greater than half the amplitude of the larger peak. Spatial boundaries were calculated as the space bins at the left and right edges of spatial footprints. To identify spatial regions with statistically significant spatial tuning, we employed a localized information metric. First, for each peak identified in the tuning curves, we extended its footprint by half its width in both directions. Within these extended regions, a local Skaggs' information metric was calculated. To assess the significance of this metric, a null distribution was generated using bootstrapping as above (1500 iterations). The inferred activity for the selected region was circularly permuted with a random time shift in each lap, followed by recalculating the Skaggs' information content. The 95th percentile of this null distribution served as the significance criterion to identify regions exhibiting statistically high spatial location selectivity. Regions with statistically high spatial selectivity were considered as PFs if they had at least one inferred activity in at least 30% of the laps.

### Temporal demarcation

To ensure a conservative estimate of PFF time and avoid misidentification of small neuronal activities or noise as formation events, the inferred activity data below 64 a.u. was set to zero. This step excluded potentially ambiguous neuronal activities that might be difficult to confirm visually in the motion-corrected videos (see in Quality control of signal isolation for BTSP- and non-BTSP-related PFFs).

For analyzing the formation and end laps of newly formed PFs, we used this thresholded inferred activity data, within the previously identified PF spatial boundaries but without filtering based on the running speed of the animal. To account for the potential for PFs to exhibit a backward shift in activity relative to the formation lap, we extended the previously identified spatial boundaries in the forward direction (to the right) by 5 space bins or until the end of the corridor. This approach ensured that formation events occurring at locations forward compared to the PF's spatial center (as expected for BTSP) would be detected. We identified potential PFF events by analyzing the inferred activity data across laps. First, we constructed a PF activity matrix by calculating the average activity for each space bin within the PF across all laps. To ensure minimal pre-formation event activity, only laps after the 17th were considered for potential formation events. A lap was classified as a formation lap only if no more than 10% of the previous laps had a mean activity exceeding 1% of the mean activity of the potential formation lap. In maze extension experiments (*Figure 4*), the minimum 17 pre-formation laps requirement was ignored. A PF was considered to be terminated if at least five consecutive laps with no activity occurred or the final lap of the session was reached.

## Categorization of newly formed PFs

We employed a three-step process to categorize newly formed PFs. First, we excluded PFs that were formed in the first 17 laps. Second, we excluded PFs that exhibited a lifetime <9 laps. Finally, we addressed activity consistency within PFs. Lap-wise maximum activity rates of the PF activity matrix were normalized to the peak activity rate across all laps. PFs were excluded if the number of silent laps (laps with normalized activity below 10%) exceeded eight laps. PFs were categorized as showing formation lap gain if the peak activity value within the spatial bounds of the PF was greater in the formation lap than the average peak activity value observed across the subsequent three active laps. For further categorization of the newly formed PFs, the location of the COM was calculated for each active lap and smoothed across laps using a running average with a kernel size of 3. The formation location of the PF was defined as the activity COM in the formation lap. Finally, the mean COM location for the first three post-formation active laps was calculated. If this mean value was smaller than the formation location value, the PF was categorized as exhibiting an initial backward shift. To exclude misinterpretation of occasionally observed continuous drift of the PF COM as BTSP-like initial shift,

we also quantified continuous backward drift (termed 'drift') in the PF. For this, we first isolated the smoothed COM location values for each lap, starting from the fourth active lap after the formation. We then performed a linear regression analysis on these COM location value series. If the calculated COM location value in the first lap after the formation lap extrapolated from the regression line was not larger than the formation location or the Spearman's rank correlation coefficient between the COM location values and their sequential labels was not statistically significant ($p < 0.05$) with a negative value, we categorized the PF as lacking significant drift.

We categorized PFF events based on their activity patterns during and after formation. PFFs were considered BTSP-like if they fulfilled all of the following criteria: (1) showed formation lap gain, (2) followed by a backward shift, and (3) showed no significant further drift. Conversely, PFFs that lacked both (1) formation lap gain and (2) an initial backward shift during formation, regardless of the PFs' drift characteristics, were classified as non-BTSP-like. These candidate BTSP- and non-BTSP-like PFFs were further analyzed as detailed below (see Quality control of signal isolation for BTSP- and non-BTSP-related PFFs). In our initial categorization of PFFs detected in the standard maze (*Figures 1 and 2*), we found 806 candidate BTSP-like (52%) and 164 non-BTSP-like (10%) PFFs; an additional 593 PFs (38%) could not be included in the above two groups [40 PFs (3%) with formation lap gain and backward shift but significant backward drift; 238 PFs (15%) with formation lap gain but without backward shift; 315 PFs (20%) with no formation lap gain but with backward shift]; these PFs were excluded from further analysis.

We emphasize that we employed a conservative approach for categorizing PFFs: we only included PFF events in our analysis if they fulfilled all criteria of either BTSP- or non-BTSP-like groups and excluded PFFs that failed on at least one criterion. Yet, it should be noted that BTSP might occur without positive gain (if the cell fires CSBs in the PF on multiple laps after the formation lap) and possibly without negative shift (*Madar et al., 2025*).

## Quality control of signal isolation for BTSP- and non-BTSP-related PFFs

Automated ROI segmentation by Suite2p can encompass processes from multiple cells which can lead to erroneously reported $[Ca^{2+}]$ dynamics during PF activity. To address this and facilitate human confirmation of automated ROI segmentation, we isolated short motion-corrected video segments from place cells and their surroundings exhibiting BTSP- and non-BTSP-like PFFs. To quantify the spatial footprints of active neurons during the animal's PF transitions, we employed a method to assess local coherence of fluorescence signals within the field of view. For each pixel in an isolated video frame, we assigned an intensity by averaging the Pearson's product-moment correlation coefficients between its time series and the time series of its eight nearest neighbors. In the resultant grayscale image, pixels corresponding to active cells were given higher intensity values, reflecting the synchronized activation of these pixels during neuronal spiking. The contrast of the resulting grayscale image was further enhanced with Scikit-image's exposure.adjust_sigmoid function (cutoff = 0.8, gain = 20). Finally, to incorporate the temporal aspect of pixel activation patterns, we applied a pseudo-color map. This map assigned a color to each pixel based on the COM of its intensity time series. In essence, these spatial footprints showed the shape of active cells with their subregions colored based on their average timing of activity. The spatial footprints of each PF transition were visually inspected and ROIs with multicolored subregions were excluded. This analysis was performed for all candidate BTSP- and non-BTSP-like PFFs in all experiments (normal maze, extended maze, and enriched maze).

## Estimating the probability that a non-BTSP-like PFF event is misclassified

To assess the likelihood that a non-BTSP PFF is the consequence of random occurrence of a small $[Ca^{2+}]$ transient before a BTSP-like PFF event in the future PF, we sequentially masked laps starting from the formation lap of a non-BTSP event until a BTSP signature was detected or the PF ended. If no BTSP signature was found, the transformation probability was set to zero. Otherwise, we defined a test area between the non-BTSP event start lap and the first BTSP-like event formation lap. Within this test area, we calculated the following two metrics: mean activity and the fraction of active laps. To generate a null distribution, we randomly sampled equal-sized segments from areas outside PFs and calculated the same metrics. A non-BTSP PFF was considered unlikely to be a misclassified BTSP PFF

due to random activity if fewer than 5% of the null samples had a larger or equal fraction of active laps and a larger or equal mean activity.

## Detection of solitary [Ca²⁺] transients

Solitary [$Ca^{2+}$] transients were identified by the following process. First, for each BTSP-like PFF event, we calculated the maximum $\Delta F/F$ value by analyzing time points recorded during the animal's traversal across the width of the PF in the formation lap, extended by five space bins to the right but not further than the end of the corridor. This maximum amplitude served as the parameter for peak detection (Scipy, 'find_peaks', prominence = 1, distance = 20) to identify peaks equal to or larger than this formation lap peak amplitude. Second, [$Ca^{2+}$] transients within the spatial boundaries of any existing or newly formed PFs were excluded. To further ensure that the detected solitary peak was not part of a short-lived place-modulated activity, we calculated the maximum $\Delta F/F$ value from the previous and following 5 laps within 11 spatial bins centered around the peak-containing space bin. If the maximum $\Delta F/F$ in this area exceeded the median plus two SD of the ROIs' $\Delta F/F$ values from the entire session, the solitary event was excluded. For each ROI, only the largest solitary peak was used for calculating the mean waveform, but we note that in many cells, multiple [$Ca^{2+}$] transients in a given session fulfilled the criteria for solitary events.

## Lap-by-lap PV correlations

For each Suite2p ROI, the non-thresholded inferred mean activity data collected during each lap in each space bin transition was computed. Subsequently, Pearson correlation coefficients were calculated between the activities of all the ROIs at each spatial bin for every possible lap pair. Finally, the lap-by-lap PV correlation for a given lap pair was determined by averaging these location-specific correlation coefficients across spatial bins.

## Odd-even lap PV correlations

To evaluate population-wide activity changes upon the extension of the familiar corridor by a new maze segment, we computed PV correlations between odd and even laps for the first 6 meters of the familiar pre-extension maze (excluding the RZ) and the 6-m-long new extension part of the maze. For each ROI, we first calculated the mean non-thresholded inferred activity from each spatial bin for both odd and even laps from the session immediately preceding novel maze segment insertion and during the extended maze experiment. These PVs were then used to calculate the average pairwise Pearson correlations between intra-corridor and cross-corridor odd–even lap data for each animal. Finally, we compared the means of these correlations across animals for both intra-corridor and cross-corridor PV correlations.

## Dependence of PF width on the running speed and cue distance

For the analysis in *Figure 3B–D*, PFs were excluded if their boundaries intersected with the start or end of the corridor, ensuring that PF widths were not truncated. In addition, PFF events where the running speed was lower than 5 cm/s were excluded.

We determined centroid coordinates of the discrete visual landmarks and calculated the Euclidean distance to the nearest landmark centroid for each point within the virtual maze.

## Variance explained by speed and cue distance

We employed leave-one-out cross-validation and standard linear regression (statsmodels GLM, Gaussian family, identity link function) to model PF widths. Predictor variables were mean running speed during PF traversal on the formation lap and distance to the nearest landmark centroid or both. For each cross-validation iteration, the fitted model was used to predict the width of the left-out PF and prediction performance was evaluated using mean squared error (MSE). To quantify the proportion of variance explained, MSE values were normalized to the variance of the PF widths and the explained variance fraction was calculated as 1 − normalized MSE.

The animal's subjective location within the VR and its distance from any virtual cues may be ambiguous. To address this, we iteratively tested starting positions within the first 50 cm of the VR corridor, and we selected the starting location in each animal that maximized the explained variance of PF

width by the combined speed and cue distance model. This optimized starting position was used for the reported explained variance values in *Figure 3D*.

## Comparison of inferred spikes and fluorescence

To compare PFs identified using inferred spike- and normalized fluorescence ($\Delta F/F$)-based raster plots, we focused on ROIs exhibiting at least one newly formed PF in the regular maze sessions (*Figure 3—figure supplement 1C–E*). For each cell, we used data from the previously determined spatial footprints of both newly formed and existing PFs. Within each ROI, we concatenated the data from all identified PFs along the laps dimension of the raster plots. We then extracted the maximum $\Delta F/F$ value and the maximum of bin-wise mean inferred spike counts for each lap within this concatenated sequence. Subsequently, we calculated the Pearson correlation coefficient between the pairs of maximum $\Delta F/F$ values and maximum mean inferred spike counts for each ROI. To generate a null distribution for comparison, we performed all possible lap-wise circular permutations of the inferred spike count values within the concatenated sequence. This effectively decoupled the relationship between spike counts and the fluorescence. For each permutation, we calculated the Pearson correlation coefficients.

## Statistics

Statistical analyses were performed using Statistica (general linear model with repeated measures or mixed ANOVA; software version 14.0.1.25, Tibco) or Python (correlations, bootstrapping, explained variances). Tukey's test was applied for post hoc comparisons after a significant effect was demonstrated by the main ANOVA. In all figures, statistical significance is indicated as $*p < 0.05$, $**p < 0.01$, $***p < 0.001$, n.s.: not significant.

## Acknowledgements

ZN is the recipient of a European Research Council Advanced Grant (ERC-AG 787157) and a National Research and Innovation Office Advanced grant (ADVANCED 150798). JKM was supported by a European Research Council Consolidator Grant (ERC-CoG 771849) and the International Research Scholar Program of the Howard Hughes Medical Institute (55008740). The work of JKM and ZN is supported by the National Brain Research Program (NAP 3.0) of the Hungarian Academy of Sciences (NAP2022-I-1/2022). The financial support from these funding bodies is gratefully acknowledged. We thank Balázs Ujfalussy for helpful discussions and comments on the manuscript.

## Additional information

### Funding

| Funder | Grant reference number | Author |
| --- | --- | --- |
| European Research Council | 10.3030/787157 | Zoltan Nusser |
| European Research Council | 10.3030/771849 | Judit K Makara |
| Hungarian Science Foundation | ADVANCED 150798 | Zoltan Nusser |
| Howard Hughes Medical Institute | 55008740 | Judit K Makara |
| Hungarian Academy of Sciences | NAP2022-I-1/2022 | Judit K Makara Zoltan Nusser |

The funders had no role in study design, data collection, and interpretation, or the decision to submit the work for publication.

## Author contributions
Mate Sumegi, Data curation, Software, Formal analysis, Investigation, Visualization, Methodology, Writing – review and editing; Gaspar Olah, Istvan Paul Lukacs, Data curation, Investigation, Writing – review and editing; Martin Blazsek, Software, Methodology, Writing – review and editing; Judit K Makara, Conceptualization, Supervision, Funding acquisition, Validation, Methodology, Writing – original draft, Project administration; Zoltan Nusser, Conceptualization, Supervision, Funding acquisition, Validation, Writing – original draft, Project administration

## Author ORCIDs
Mate Sumegi https://orcid.org/0009-0009-3600-2570
Gaspar Olah http://orcid.org/0000-0003-4708-2368
Istvan Paul Lukacs http://orcid.org/0000-0003-0257-5441
Judit K Makara https://orcid.org/0000-0001-8134-6334
Zoltan Nusser https://orcid.org/0000-0001-7004-4111

## Ethics
All procedures adhered to the regulations outlined in the Hungarian Act of Animal Care and Experimentation (40/2013 [II.14]) and received approval from the Animal Committee of the Institute of Experimental Medicine, Budapest.

Reviewer #1 (Public review): https://doi.org/10.7554/eLife.103676.3.sa1
Reviewer #2 (Public review): https://doi.org/10.7554/eLife.103676.3.sa2
Reviewer #3 (Public review): https://doi.org/10.7554/eLife.103676.3.sa3
Author response https://doi.org/10.7554/eLife.103676.3.sa4

# Additional files

## Supplementary files
MDAR checklist

## Data availability
Data and analysis code of this manuscript are available at https://hdl.handle.net/21.15109/ARP/YGNZNJ.

The following dataset was generated:

| Author(s) | Year | Dataset title | Dataset URL | Database and Identifier |
|---|---|---|---|---|
| Sumegi M, Olah G, Lukacs IP, Blazsek M, Makara KJ, Nusser Z | 2025 | Diverse calcium dynamics underlie place field formation in hippocampal CA1 pyramidal cells | https://hdl.handle.net/21.15109/ARP/YGNZNJ | The ARP Research Data Repository of the Hungarian Research Network, 21.15109/ARP/YGNZNJ |

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
