## [Editor Report · eLife Assessment]

This **fundamental** study provides new insights into the plasticity mechanisms underlying the formation of spatial maps in the hippocampus. Supported by a large and comprehensive dataset, the evidence is **convincing**. This study will be of interest to neuroscientists focusing on spatial navigation, learning, and memory.

---

## [Referee Report · Reviewer #1 (Public review)]

Summary:

The authors aimed to investigate the cellular mechanisms underlying place field formation (PFF) in hippocampal CA1 pyramidal cells by performing in vivo two-photon calcium imaging in head-restrained mice navigating a virtual environment. Specifically, they sought to determine whether BTSP-like (behavioral time scale synaptic plasticity) events, characterized by large calcium transients, are the primary mechanism driving PFFs or if other mechanisms also play a significant role. Through their extensive imaging dataset, the authors found that while BTSP-like events are prevalent, a substantial fraction of new place fields are formed via non-BTSP-like mechanisms. They further observed that large calcium transients, often associated with BTSP-like events, are not sufficient to induce new place fields, indicating the presence of additional regulatory factors (possibly local dendritic spikes).

Strengths

The study makes use of a robust and extensive dataset collected from 163 imaging sessions across 45 mice, providing a comprehensive examination of CA1 place cell activity during navigation in both familiar and novel virtual environments. The use of two-photon calcium imaging allows the authors to observe the detailed dynamics of neuronal activity and calcium transients, offering insights into the differences between BTSP-like and non-BTSP-like PFF events. The study's ability to distinguish between these two mechanisms and analyze their prevalence under different conditions is a key strength, as it provides a nuanced understanding of how place fields are formed and maintained. The paper supports the idea that BTSP is not the only driving fore behind PFF, and other mechanisms are likely sufficient to drive PFF, and BTSP events may also be insufficient to drive PFF in some cases. The longer-than-usual virtual track used in the experiment allowed place cells to express multiple place fields, adding a valuable dimension to the dataset that is typically lacking in similar studies. Additionally, the authors took a conservative approach in classifying PFF events, ensuring that their findings were not confounded by noise or ambiguous activity.

Weaknesses

The stand out weakness of the paper is the lack of direct measures of BTSP events. Without direct confirmation that large calcium transients correspond to actual BTSP events (including associated complex spikes and calcium plateau potentials), concluding that BTSP is not necessary or sufficient for PFF formation is speculative (although I do believe it).

---

## [Referee Report · Reviewer #2 (Public review)]

Summary:

The authors of this manuscript aim to investigate the formation of place fields (PFs) in hippocampal CA1 pyramidal cells. They focus on the role of behavioral time scale synaptic plasticity (BTSP), a mechanism proposed to be crucial for the formation of new PFs. Using in vivo two-photon calcium imaging in head-restrained mice navigating virtual environments, employing a classification method based on calcium activity to categorize the formation of place cells' place fields into BTSP, non-BTSP-like, and investigated their properties.

Strengths:

This work shows that place fields formation could induced by both BSTP and non-BSTP events, and it also provided a new and solid method to classify BTSP and non-BTSP place field formation using calcium image to the field. This work offers novel knowledge and new methods and factual evidence for other researchers in the field.

The method enabled the authors to reveal that while many PFs are formed by BTSP-like events, a significant number of PFs emerge with calcium dynamics that do not match BTSP characteristics, suggesting a diversity of mechanisms underlying PF formation. The characteristics of place fields under the first two categories are comprehensively described, including aspects such as formation timing, quantity, and width.

Weaknesses:

The authors have addressed the weaknesses in the revised version.

---

## [Referee Report · Reviewer #3 (Public review)]

Summary:

In this manuscript, Sumegi et al. use calcium imaging in head-fixed mice to test whether new place fields tend to emerge due to events that resemble behavioral time scale plasticity (BTSP) or other mechanisms. An impressive dataset was amassed (163 sessions from 45 mice with 500-1000 neurons per sample) to study spontaneous emergence of new place fields in area CA1 that had the signature of BTSP. The authors observed that place fields could emerge due to BTSP and non-BTSP-like mechanisms. Interestingly, when non-BTSP mechanisms seemed to generate a place field, this tended to occur on a trial with a spontaneous reset in neural coding (a remapping event). Novelty seemed to upregulate non-BTSP events relative to BTSP events. Finally, large calcium transients (presumed plateau potentials) were not sufficient to generate a place field.

Strengths:

I found this manuscript to be exceptionally well written, well powered, and timely given the outstanding debate and confusion surrounding whether all place fields must arise from BTSP event. Working at the same institute, Albert Lee (e.g. Epszstein et al., 2011 - which should be cited) and Jeff Magee (e.g. Bittner et al., 2017) showed contradictory results for how place fields arise. These accounts have not fully been put toe-to-toe and reconciled in the literature. This manuscript addresses this gap and shows that both accounts are correct - place fields can emerge due to a pre-existing map and due to BTSP.

Weaknesses:

I find only three significant areas for improvement in the present study:

First, can it be concluded that non-BTSP events occur exclusively due to a global remapping event, as stated in the manuscript "these PFF surges included a high fraction of both non-BTSP- and BTSP-like PFF events, and were associated with global remapping of the CA1 representation"? Global remapping has a precise definition that involves quantifying the stability of all place fields recorded. Without a color scale bar in Figure 3D (which should be added), we cannot know whether the overall representations were independent before and after the spontaneous reset. It would be good to know if some neurons are able to maintain place coding (more often than expected by chance), suggestive of a partial-remapping phenomenon.

Second, BTSP has a flip side that involves weakening of existing place fields when a novel field emerges. Was this observed in the present study? Presumably place fields can disappear due to this bidirectional-BTSP or due to global remapping. For a full comparison of the two phenomena, the disappearance of place fields must also be assessed.

Finally, it would be good to know if place fields differ according to how they are born. For example, are there differences in reliability, width, peak rate, out of field firing, etc for those that arise due BTSP vs non-BTSP.

Comments on revisions:

The authors have mostly addressed my feedback. Compelling evidence for a fundamental observation.

---

## [Author Response]

The following is the authors’ response to the original reviews.

**Reviewer #1 (Public review):**
Summary:The authors aimed to investigate the cellular mechanisms underlying place field formation (PFF) in hippocampal CA1 pyramidal cells by performing in vivo two-photon calcium imaging in head-restrained mice navigating a virtual environment. Specifically, they sought to determine whether BTSP-like (behavioral time scale synaptic plasticity) events, characterized by large calcium transients, are the primary mechanism driving PFFs or if other mechanisms also play a significant role. Through their extensive imaging dataset, the authors found that while BTSP-like events are prevalent, a substantial fraction of new place fields are formed via non-BTSP-like mechanisms. They further observed that large calcium transients, often associated with BTSP-like events, are not sufficient to induce new place fields, indicating the presence of additional regulatory factors (possibly local dendritic spikes).StrengthsThe study makes use of a robust and extensive dataset collected from 163 imaging sessions across 45 mice, providing a comprehensive examination of CA1 place-cell activity during navigation in both familiar and novel virtual environments. The use of two-photon calcium imaging allows the authors to observe the detailed dynamics of neuronal activity and calcium transients, offering insights into the differences between BTSP-like and non-BTSP-like PFF events. The study's ability to distinguish between these two mechanisms and analyze their prevalence under different conditions is a key strength, as it provides a nuanced understanding of how place fields are formed and maintained. The paper supports the idea that BTSP is not the only driving force behind PFF, and other mechanisms are likely sufficient to drive PFF, and BTSP events may also be insufficient to drive PFF in some cases. The longer-than-usual virtual track used in the experiment allowed place cells to express multiple place fields, adding a valuable dimension to the dataset that is typically lacking in similar studies. Additionally, the authors took a conservative approach in classifying PFF events, ensuring that their findings were not confounded by noise or ambiguous activity.WeaknessesDespite the impressive dataset, there are several methodological and interpretational concerns that limit the impact of the findings. Firstly, the virtual environment appears to be poorly enriched, relying mainly on wall patterns for visual cues, which raises questions about the generalizability of the results to more enriched environments. Prior studies have shown that environmental enrichment can significantly influence spatial coding, and it would be important to determine how a more immersive VR environment might alter the observed PFF dynamics. Secondly, the study relies on deconvolution methods in some cases to infer spiking activity from calcium signals without in vivo ground truth validation. This introduces potential inaccuracies, as deconvolution is an estimate rather than a direct measure of spiking, and any conclusions drawn from these inferred signals should be interpreted with caution. Thirdly, the figures would benefit from clearer statistical annotations and visual enhancements. For example, several plots lack indicators of statistical significance, making it difficult for readers to assess the robustness of the findings. Furthermore, the use of bar plots without displaying underlying data distributions obscures variability, which could be better visualized with violin plots or individual data points. The manuscript would also benefit from a more explicit breakdown of the proportion of place fields categorized as BTSP-like versus non-BTSP-like, along with clearer references to figures throughout the results section. Lastly, the authors' interpretation of their data, particularly regarding the sufficiency of large calcium transients for PFF induction, needs to be more cautious. Without direct confirmation that these transients correspond to actual BTSP events (including associated complex spikes and calcium plateau potentials), concluding that BTSP is not necessary or sufficient for PFF formation is speculative.
**Reviewer #2 (Public review):**
Summary:The authors of this manuscript aim to investigate the formation of place fields (PFs) in hippocampal CA1 pyramidal cells. They focus on the role of behavioral time scale synaptic plasticity (BTSP), a mechanism proposed to be crucial for the formation of new PFs. Using in vivo two-photon calcium imaging in head-restrained mice navigating virtual environments, employing a classification method based on calcium activity to categorize the formation of place cells' place fields into BTSP, non-BTSP-like, and investigated their properties.Strengths:A new method to use calcium imaging to separate BTSP and non-BTSP place field formation. This work offers new methods and factual evidence for other researchers in the field.The method enabled the authors to reveal that while many PFs are formed by BTSP-like events, a significant number of PFs emerge with calcium dynamics that do not match BTSP characteristics, suggesting a diversity of mechanisms underlying PF formation. The characteristics of place fields under the first two categories are comprehensively described, including aspects such as formation timing, quantity, and width.Weaknesses:There are some issues about data and statistics that need to be addressed before these research findings can be considered as rigorous conclusions.While the authors mentioned 3 features of PF generated by BTSP during calcium imaging in the Introduction, the classification method used features 1 and 2. The confirmation by feature 3 in its current form is important but not strong enough.Some key data is missing such as the excluded PFs, the BTSP/non-BTSP of each animal, etcImpact:This work is likely to provide a new method to classify BTSP and non-BTSP place field formation using calsium image to the field.
**Reviewer #3 (Public review):**
Summary:In this manuscript, Sumegi et al. use calcium imaging in head-fixed mice to test whether new place fields tend to emerge due to events that resemble behavioral time scale plasticity (BTSP) or other mechanisms. An impressive dataset was amassed (163 sessions from 45 mice with 500-1000 neurons per sample) to study the spontaneous emergence of new place fields in area CA1 that had the signature of BTSP. The authors observed that place fields could emerge due to BTSP and non-BTSP-like mechanisms. Interestingly, when non-BTSP mechanisms seemed to generate a place field, this tended to occur on a trial with a spontaneous reset in neural coding (a remapping event). Novelty seemed to upregulate non-BTSP events relative to BTSP events. Finally, large calcium transients (presumed plateau potentials) were not sufficient to generate a place field.Strengths:I found this manuscript to be exceptionally well-written, well-powered, and timely given the outstanding debate and confusion surrounding whether all place fields must arise from BTSP event. Working at the same institute, Albert Lee (e.g. Epszstein et al., 2011 - which should be cited) and Jeff Magee (e.g. Bittner et al., 2017) showed contradictory results for how place fields arise. These accounts have not fully been put toe-to-toe and reconciled in the literature. This manuscript addresses this gap and shows that both accounts are correct - place fields can emerge due to a pre-existing map and due to BTSP.

We thank the Reviewer for his/her appreciation of the importance of our study. We have included the additional reference.

Weaknesses:I find only three significant areas for improvement in the present study:First, can it be concluded that non-BTSP events occur exclusively due to a global remapping event, as stated in the manuscript "these PFF surges included a high fraction of both non-BTSP- and BTSP-like PFF events, and were associated with global remapping of the CA1 representation"? Global remapping has a precise definition that involves quantifying the stability of all place fields recorded. Without a color scale bar in Figure 3D (which should be added), we cannot know whether the overall representations were independent before and after the spontaneous reset. It would be good to know if some neurons are able to maintain place coding (more often than expected by chance), suggestive of a partial-remapping phenomenon.

We have performed the analysis suggested by the Reviewer and determined what fraction of CA1PCs retained its original tuning property after the representation switch. We found that the remapping was essentially global, as only a small fraction (5.4%) of CA1PCs retained their pre-switch tuning curve after the switch. This is now described in the Results.

We now state in the figure legend for the former Figure 3D (now Figure 3F) that the color scale applies to all subpanels.

We would like to note that we do not conclude that non-BTSP events occur exclusively during global remapping – we have found a sizable fraction of PFF by non-BTSP mechanism also in the familiar environment with no signs of change in the population representation. We agree nonetheless that PFF is dominated by BTSP under these conditions, whereas the contribution of non-BTSP is larger during global remapping events.

Second, BTSP has a flip side that involves the weakening of existing place fields when a novel field emerges. Was this observed in the present study? Presumably place fields can disappear due to this bidirectional BTSP or due to global remapping. For a full comparison of the two phenomena, the disappearance of place fields must also be assessed.

In this study we focused on the birth of new PFs – yet, PFs not only form but also disappear constantly. The factors driving PF weakening are even less explored and understood than those driving PF birth. In fact, we observed (as illustrated by several examples in our MS) that many PFs weaken, or disappear completely during the course of an imaging session. These effects are sometimes accompanied by a new PFF event elsewhere (e.g. Figure 2 – figure supplement 2E bottom), whereas in other cases they are not (e.g. Figure 5A, middle). Similarly, some BTSP events seem to coincide with disappearance of another PF, but others are not (e.g. Figure 2A bottom, first PF along the track; Figure 3 – figure supplement 1A left, first PF). The picture is further complicated in the case of global remapping events (i.e. representation switches, Figure 3 – figure supplement 2B) that, by definition, include both new PFF and PF disappearance. We feel that exploration of the complex mechanisms at play in PF disappearance is outside the scope of the current study, but could be the subject of an interesting future investigation.

Finally, it would be good to know if place fields differ according to how they are born. For example, are there differences in reliability, width, peak rate, out-of-field firing, etc for those that arise due to BTSP vs non-BTSP.

We have analyzed several properties of the PFs and found no significant difference in either their width (BTSP: 46.4 ± 24.4 cm; non-BTSP: 50.4 ± 32.5 cm, p = 0.28) or peak rates (BTSP: 19.0 ± 14.7 a.u./s; non-BTSP: 21.4 ± 16.8 a.u./s, p = 0.27) or the out-of-field firing rates (BTSP: 0.64 ± 0.68 a.u./s; non-BTSP: 0.83 ± 1.25 a.u./s, p = 0.09, all unpaired t-test). We have included these data into the Results section.

**Reviewer #1 (Recommendations for the authors):**
Consider adding additional visual cues or environmental elements to the virtual reality (VR) setup to create a more enriched and immersive environment. Collect data from a couple of mice in the enriched environment and compare the PFF dynamics to the original environment. This would help determine whether the findings on PFF dynamics hold in a setting where spatial coding may be more robust. Including floor cues, distal visual markers, or varying textures might provide a more comprehensive understanding of the factors influencing BTSP-like and non-BTSP-like events.

We thank the Reviewer for her/his suggestion of analyzing data obtained from a more enriched VR environment compared to the one we used in our study. We have now included data obtained in a profoundly different VR environment, which did not have sparse dominant visual landmarks, but the entire wall was covered with a rich pattern with different shapes of different colors. Our data from 11 imaging sessions from 4 mice revealed BTSP- and non-BTSP-like PFF events with approximately the same ratio to that found in our regular maze. These results are described in the Results section and are presented in a new supplementary figure (Figure 2 – figure supplement 2).

Wherever deconvolved spikes were used for analysis, provide a comparison of results obtained directly from the GCaMP ΔF/F signals versus those derived from the deconvolved spiking data. This could illustrate any differences and help readers understand the limitations and reliability of the inference method.

We have adopted a currently widely accepted method in the field to infer spikes from fluorescent traces using the Suite2p software package. All of our analyses were then performed on the inferred spikes. To address the concerns of the Reviewer, we analyzed the relationship between the peak [Ca^2+^] transients and inferred spike activity (new Figure 3 – figure supplement 1C-E). Our results clearly demonstrate a robust, highly significant correlation between these measures at the level of individual cells (new Figure 3 – figure supplement 1D) and the Spearman correlation coefficients show a distribution that is very different from random distributions (new Figure 3 – figure supplement 1E). From these, we conclude that using directly the fluorescent data would have resulted in largely similar PF detection and identification.

Improve the visual clarity of figures by enlarging key elements such as arrows that indicate BTSP-like events. Consider using colors that stand out more clearly to guide readers' attention. Include annotations of statistical significance directly on the figures (e.g., adding NS or * indicators) to make it clear which comparisons are statistically significant. This will help readers quickly interpret the data without needing to refer back to the text.

Based on the suggestion of the Reviewer, we have enlarged the arrows. We have also indicated statistical results on the figures. Because some of the results of factorial ANOVA tests are difficult to be comprehensively indicated on our plots, we kept the description of the statistical results in the legends as well. We hope that these alterations will make data interpretation easier.

Replace or supplement bar plots with violin plots or scatter plots that show the distribution of individual data points. This change would offer a clearer picture of data variability and underlying trends, aiding readers in assessing the robustness of the results.

We have changed the plots and now present all data points.

Add more detailed quantification in the results section, specifying the total number of newly formed place fields, the proportion that are categorized as BTSP-like versus non-BTSP-like, and how many events did not fit these categories. Explicitly state what fraction of the total recorded place field formations are represented by the 59 non-BTSP-like events mentioned, as this is currently difficult to discern.

The number of BTSP- and non-BTSP-like PFF events are given in the MS. As described in the Methods, after identifying BTSP- and non-BTSP-like PFF events using the shift and gain criteria, we have manually checked each of these ROIs and the spatial footprint of every new PFF events for these cells and excluded ROIs with non-soma-like shapes and activities with spurious footprints suggesting contamination, creating a ‘cleaned’ dataset. We did not perform such visual inspection and manual curation of every ROI’s spatial footprints that belong to the two additional categories (no gain with shift, gain without shift, 872 events). Since these classes are also overestimated without curation, we cannot provide a precise fraction of the BTSP- and non-BTSP-like PFF events from the total recorded PFF population. However, - assuming that factors leading to exclusion affect all groups equally - we can provide their fractions by comparing the numbers of newly born PFs in all categories before the visual inspections. In the normal maze, we found 806 candidate BTSP-like (52%),164 non-BTSP-like (10%) PFFs and an additional 593 PFs (38%) could not be included in these two groups [40 PFs (3%) with formation lap gain and backward shift but significant backward drift; 238 PFs (15%) with formation lap gain but without backward shift; 315 PFs (20%) with no formation lap gain but with backward shift]. These data have been included in the Methods.

Ensure that all statements describing specific findings are consistently linked to the appropriate figures and panels. There are instances in the text where results are discussed without clear references, which can make it challenging for readers to verify the data. For example, the section on population remapping in a novel environment should point directly to the relevant figure panels to guide readers.

We regret that our text was not linked properly to the appropriate figures. We corrected this during the revision.

Given that BTSP-like events are inferred rather than directly confirmed, it would be prudent to frame conclusions about their sufficiency in more tentative terms, acknowledging the limitations of the current data. Consider adding a discussion of potential future experiments that could confirm whether these large transients truly represent BTSP events, including evidence for complex spikes or calcium plateau potentials.

The Reviewer is correct that we do not have direct evidence that all large somatic Ca^2+^ events represent dendritic plateau potentials. Now we discuss this and other limitations in the MS (Discussion section).

**Reviewer #2 (Recommendations for the authors):**
Although the author has outlined three characteristics of place fields (PFs) generated by behavioral time scale synaptic plasticity (BTSP) during calcium imaging in the Introduction section, as follows: ' First, the prolonged CSB results in large [Ca^2+^] transient during the initial PFF event, typically followed by weaker Ca2+ signals on consecutive traversals through the PF. Second, due to the long and asymmetric temporal kernel of the plasticity (favoring potentiation of inputs active 1-2 seconds before the CSB) a substantial backward shift in the spatial position of the PF center can be observed on linear tracks after the formation lap. Third, the width of the new PF is generally proportional to the running speed of the animal during the PFF event.' Figure 3B, which displays the third feature of classified BTSP and non-BTSP data, serves as an important confirmation of the classification results using the first two features. Even though the Spearman correlation indicated a significant difference, the raw data distributions of BTSP and non-BTSP appear similar, suggesting that a distribution of bootstrap and more stringent confirmation should be conducted to be convincing.

As described in the MS, because of the difference in the number of events in the two groups, we randomly subsampled the BTSP-like events to the sample size of the non-BTSP-like PFF events 10000 times and performed regression analysis. This bootstrapping revealed that both the r and p values of the fit to the non-BTSP data fell outside the 95% confidence interval of the bootstrapped BTSP values, indicating that the difference between the groups was robust.

In further analysis during the revision, we found that the PF width variance explained by distance from landmarks is substantially larger than the variance explained by the running speed during the formation lap. We performed a cross-validated analysis by these two factors (Figure 3D), which highlights that speed explains some of the PF width variance of BTSP-like PFFs, but none of the non-BTSP PFFs.

The proportions of the three types should be provided. page 6: ' Using a conservative approach, we categorized a new PF to be formed by a BTSP-like mechanism if it had both positive gain and negative shift values (Figure 2A; n = 310 new PFs), whereas new PFs exhibiting neither positive gain nor negative shift were considered as non-BTSP-like events (Figure 2B; n = 59). All other newly formed PFs (no-gain with backward shift and gain without backward shift) were excluded from further analysis.' The number of excluded newly formed PFs should be disclosed, as well as the distribution ratio of these three types in each animal.

The number of BTSP- and non-BTSP-like PFF events are given in the MS. As described in the Methods, after identifying BTSP- and non-BTSP-like PFF events using the shift and gain criteria, we have manually checked each of these ROIs and the spatial footprint of every new PFF events for these cells and excluded ROIs with non-soma-like shapes or spurious activities, creating a ‘cleaned’ dataset. We did not perform such visual inspection and manual curation of every ROI’s spatial footprints that belonged to the two additional categories (no gain with shift, gain without shift, 872 events). Since these classes are also overestimated without curation, we cannot provide a precise fraction of the BTSP- and non-BTSP-like PFF events from the total recorded PFF population. However, - assuming that factors leading to exclusion affect all groups equally - we can provide their fractions by comparing the numbers of newly born PFs in all categories before the visual inspections. In the normal maze, we found 806 candidate BTSP-like (52%),164 non-BTSP-like (10%) PFFs and an additional 593 PFs (38%) could not be included in these two groups [40 PFs (3%) with formation lap gain and backward shift but significant backward drift; 238 PFs (15%) with formation lap gain but without backward shift; 315 PFs (20%) with no formation lap gain but with backward shift]. These data have been included in the Methods.

Figure 2C, while showing an overall decrease in amplitude from the formation lap to the next lap, could benefit from a pairwise analysis of the corresponding formation lap and the following lap of each session to provide more convincing and detailed results.

We now present all data with connected lines across consecutive laps to illustrate the changes in each ROI. Our statistical analysis included the pairwise comparison of amplitudes.

The experiment's time range is broad (11-99 days); it is worth investigating whether different training intervals might influence the results.

Based on the suggestion of the Reviewer, we have analyzed the elapsed time and the number of sessions from the first training to the recording, and we demonstrate that there is no correlation of these parameters with the number of new PFFs. These data are now presented in Figure 2 – figure supplement 1C.

It is unclear whether the formation of place fields also generates characteristic features of dendritic properties.

It is not clear to us which ‘characteristic dendritic features of dendritic properties’ generated by PFF the Reviewer refers to. Since we did not image dendrites of individual CA1PCs, we have no information about dendritic properties of the neurons.

It may be necessary to add a clearer figure to illustrate the correlation between width and speed following the downsampling of non-BTSP-like events (refer to Figure 3B).

We have performed extensive additional analysis on the relationship of PF width with various behavioral factors, including the speed of the animal in the formation lap. Inspection of the PF width distributions along the track revealed a close association of PF width with the distance of the animal from the nearest visual landmark in the corridor, so that PFs close to landmarks were narrower than PFs between landmarks. We found that the PF width variance explained by distance from landmarks is substantially larger than the variance explained by the running speed during the formation lap. Nevertheless, there is a clear difference between BTSP-like and non-BTSP-like PFFs: running speed explains some variance in the case of BTSP-like PFFs, but none for non-BTSP-like PFFs.

We have included these findings into the Results section and created two new panels in Figure 3 (C, D) and Figure 3 – figure supplement 1 (A, B).

It is recommended that statistical results be labeled in the figures with n.s. or stars for better readability.

Based on the suggestion of the Reviewer, we have indicated statistical results on the figures. Because some of the results of factorial ANOVA tests are difficult to be comprehensively indicated on our plots, we kept the description of the statistical results in the legends as well. We hope that these alterations will make data interpretation easier. We hope that these alterations will make data interpretation easier.